# An improved bacterial single-cell RNA-seq reveals biofilm heterogeneity

Xiaodan Yan[1,2†], Hebin Liao[1,2,3†], Chenyi Wang[1,2], Chun Huang[1,2], Wei Zhang[1,2], Chunming Guo[4], Yingying Pu[1,2,5]*

[1]The State Key Laboratory Breeding Base of Basic Science of Stomatology & Key Laboratory of Oral Biomedicine Ministry of Education, School & Hospital of Stomatology, Medical Research Institute, Wuhan University, Wuhan, China; [2]Frontier Science Center for Immunology and Metabolism, Wuhan University, Wuhan, China; [3]Translational Medicine Research Center, North Sichuan Medical College, Nanchong, China; [4]Center for Life Sciences, School of Life Sciences, Yunnan University, Kunming, China; [5]Department of Immunology, Hubei Province Key Laboratory of Allergy and Immunology, State Key Laboratory of Virology and Medical Research Institute, Wuhan University School of Basic Medical Sciences, Wuhan, China

**\*For correspondence:**
yingyingpu@whu.edu.cn

†These authors contributed equally to this work

**Competing interest:** The authors declare that no competing interests exist.

## eLife Assessment

This work introduces an **important** new method for depleting ribosomal RNA from bacterial single-cell RNA sequencing libraries, demonstrating its applicability for studying heterogeneity in microbial biofilms. The findings provide **convincing** evidence for a distinct subpopulation of cells at the biofilm base that upregulates PdeI expression. Future studies exploring the functional relationship between PdeI and c-di-GMP levels, along with the roles of co-expressed genes within the same cluster, could further enhance the depth and impact of these conclusions.

**Abstract** In contrast to mammalian cells, bacterial cells lack mRNA polyadenylated tails, presenting a hurdle in isolating mRNA amidst the prevalent rRNA during single-cell RNA-seq. This study introduces a novel method, ribosomal RNA-derived cDNA depletion (RiboD), seamlessly integrated into the PETRI-seq technique, yielding RiboD-PETRI. This innovative approach offers a cost-effective, equipment-free, and high-throughput solution for bacterial single-cell RNA sequencing (scRNA-seq). By efficiently eliminating rRNA reads and substantially enhancing mRNA detection rates (up to 92%), our method enables precise exploration of bacterial population heterogeneity. Applying RiboD-PETRI to investigate biofilm heterogeneity, distinctive subpopulations marked by unique genes within biofilms were successfully identified. Notably, PdeI, a marker for the cell-surface attachment subpopulation, was observed to elevate cyclic diguanylate (c-di-GMP) levels, promoting persister cell formation. Thus, we address a persistent challenge in bacterial single-cell RNA-seq regarding rRNA abundance, exemplifying the utility of this method in exploring biofilm heterogeneity. Our method effectively tackles a long-standing issue in bacterial scRNA-seq: the overwhelming abundance of rRNA. This advancement significantly enhances our ability to investigate the intricate heterogeneity within biofilms at unprecedented resolution.

## Introduction

Biofilms, contributing to approximately 80% of chronic and recurrent microbial infections in the human body (*Costerton et al., 1999*), are complex microbial ecosystems characterized by a diverse array

of bacterial cells existing in various physiological states (*Costerton et al., 1999*; *Evans et al., 2023*; *Stewart and Franklin, 2008*). This heterogeneity within biofilms is influenced by multiple factors, including bacterial interactions (such as competition, symbiosis, and parasitism) that contribute to stable community structures (*Flemming et al., 2023*; *Shokeen et al., 2021*), environmental factors (like host environment, aquatic conditions, and nutrient concentrations), and spatial organization. Bacteria occupying different positions within the biofilm perform distinct roles (*Qian et al., 2022*): some mediate material exchange between cells and with the extracellular matrix, others facilitate complex communication systems between bacteria and with the host through signaling molecules, while certain bacteria participate in various energy conversion processes. This intricate division of labor not only contributes to bacterial heterogeneity but also enhances the biofilm's overall resistance to various stresses (*Momeni, 2018*). The resulting cellular and functional diversity reflects the complex nature of the biofilm ecosystem, allowing it to adapt and persist in challenging environments. However, the study of biofilms faces significant limitations, primarily stemming from challenges in investigating heterogeneity within a bacterial population (*Cheng et al., 2023*; *Spormann, 2008*). Single-cell RNA-seq emerges as a promising avenue for addressing this heterogeneity (*Shapiro et al., 2013*; *Tang et al., 2009*; *Blattman et al., 2020*; *Imdahl et al., 2020*; *Kuchina et al., 2021*; *Ma et al., 2023*; *Wang et al., 2023*; *McNulty et al., 2023*; *Lu et al., 2023*). Expending on established protocols for cell fixation and permeabilization which facilitate in-cell barcoding while avoiding cell lysis, combinatorial barcoding-based bacterial scRNA-seq techniques, such as prokaryotic expression profiling by tagging RNA in situ and sequencing (PETRI-seq) (*Blattman et al., 2020*) and microbial split-pool ligation transcriptomics (microSPLiT) (*Kuchina et al., 2021*), have been developed. Nevertheless, these methods encounter challenges in terms of low transcript recovery rates due to overwhelmingly abundant rRNA, restricting the comprehensive analysis of within-population heterogeneity. In comparison to mammalian cells (*Maynard et al., 2020*), the absence of mRNA polyadenylated tails in bacteria necessitates an alternative approach for isolating mRNA (~5%) from the significantly more abundant rRNA (~95%). Here, by integrating a ribosomal RNA-derived cDNA depletion protocol (RiboD) into a PETRI-seq, we developed RiboD-PETRI-seq that efficiently eliminates rRNA reads, thereby significantly improving mRNA detection rates and enabling exploration of within-population heterogeneity.

## Results

In the RiboD protocol, we designed a set of probe primers that spans all regions of the bacterial rRNA sequence (*Supplementary file 1*). The core principle behind our probe design is twofold: the 3'-end of the probes is reverse complementary to the r-cDNA sequences, allowing for specific recognition of r-cDNA, while the 5'-end complements a biotin-labeled universal primer. This design enables the probes to be bound to magnetic beads, facilitating the separation of r-cDNA-probe-bead complexes from the rest of the library. Following template switching and RNaseH treatment on the barcoded cDNA from lysed cells to eliminate hybridized RNA, the library of probe primers and biotin-labeled universal primers is introduced to facilitate adequate hybridization. Pre-treated Streptavidin magnetic beads are then added to the hybridized rRNA-derived cDNA. The mRNA-derived cDNA remains in the supernatant and is collected for subsequent library construction and sequencing (*Figure 1A*). To assess the efficiency of single-cell capture in RiboD-PETRI, we calculated the multiplet frequency (*Blattman et al., 2020*) using a Poisson distribution based on our sequencing results (see details in Materials and methods). The multiplet frequency for RiboD-PETRI ranges from 1.16% to 3.35% (*Supplementary file 2*), indicating the technique's capability to effectively capture transcriptomes at the single-cell level.

To assess the performance of RiboD-PETRI, we designed a comprehensive assessment of rRNA depletion efficiency under diverse physiological conditions, specifically contrasting exponential and stationary phases. This approach allows us to understand how these different growth states impact rRNA depletion efficacy. Additionally, we included a variety of bacterial species, encompassing both gram-negative and gram-positive organisms, to ensure that our findings are broadly applicable across different types of bacteria. By incorporating these variables, we aim to provide insights into the robustness and reliability of the RiboD-PETRI method in various biological contexts. The results highlight a substantial enhancement in rRNA-derived cDNA depletion, with mRNA ratio increases from 8.2% (Ctrl, the PETRI-seq we performed) to 81% (ΔΔ, RiboD-PETRI) for *E. coli* from exponential phase, from 10% (Ctrl) to 92% (ΔΔ) for *S. aureus* from stationary phase, and from 3.9% (Ctrl) to 54% (ΔΔ) for

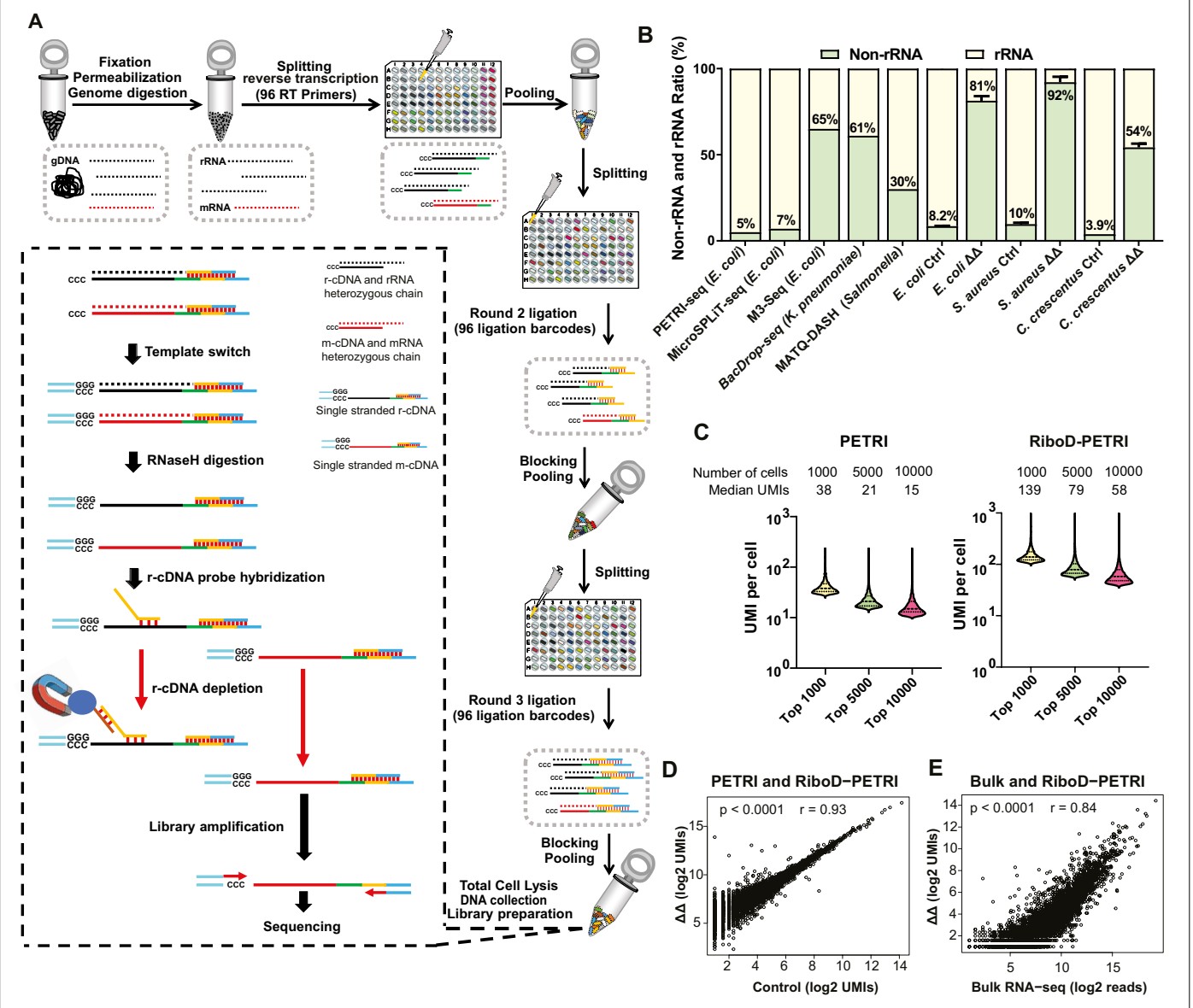

**Figure 1.** Development of RiboD-PETRI and validation of its technical performance in studying population heterogeneity. (**A**) Graphic summary of the RiboD-PETRI method illustrating the incorporation of RiboD after cell pooling and lysis in PETRI-seq. The RiboD protocol is represented by the dashed-line box. In this box, first, we perform template-switching oligonucleotides (TSOs) in the mixture of heterozygous chain, then we remove the RNA strand using RNaseH, at this point the system contains r-cDNA and m-cDNA single-stranded mixture. Then we add the r-cDNA probe, which specifically binds to the r-cDNA. The probes are then bound to magnetic beads, allowing the r-cDNA-probe-bead complexes to be separated from the rest of the library. And then we remove the r-cDNA that is attached to the probe by Streptavidin magnetic beads. We then performed amplification of the libraries and sent them for sequencing. We designed separate probe sets for *Escherichia coli*, *Caulobacter crescentus*, and *Staphylococcus aureus*. Each set was specifically constructed to be reverse complementary to the r-cDNA sequences of its respective bacterial species. This species-specific approach ensures high efficiency and specificity in rRNA depletion for each organism. (**B**) Comparison of non-rRNA (tRNA, mRNA, and other non-rRNA) and rRNA unique molecular identifier (UMI) counts ratio among different bacterial scRNA-seq methods. Data from PETRI-seq (*E. coli*), MicroSPLiT-seq (*E. coli*), M3-seq (*E. coli*) cited from previous studies. Error bars represent standard deviations of biological replicates. The 'ΔΔ' label represents the RiboD-PETRI protocol. The 'Ctrl' label represents the classic PETRI-seq protocol we performed. (**C**) Comparison of UMI counts per cell between RiboD-PETRI (*Supplementary file 7*) and PETRI (*Supplementary file 8*) at the same unsaturated sequencing depth. (**D**) Assessment of the effect of rRNA depletion on transcriptional profiles. The Pearson correlation coefficient (**r**) of UMI counts per gene (log₂ UMIs) between RiboD-PETRI (*Supplementary file 7*) and PETRI (*Supplementary file 9*) was calculated for 3790 out of 4141 total genes, excluding those with zero counts in either library. Each point represents a gene. (**E**) Evaluation of the correlation between RiboD-PETRI (*Supplementary file 7*) data and bulk RNA-seq (*Supplementary file 10*) results. The Pearson correlation coefficient (**r**) of UMI counts per gene (log₂ UMIs) among RiboD-PETRI data and the reads per gene (log₂ reads) of bulk RNA-seq

*Figure 1 continued on next page*

*Figure 1 continued*

data was calculated for 3814 out of 4141 total genes, excluding those with zero counts in either library. Each point represents a gene. All data presented in **C, D, E** were from our own sequencing experiments.

The online version of this article includes the following source data, source code, and figure supplement(s) for figure 1:

**Source code 1.** Related to *Figure 1*.

**Source data 1.** Related to *Figure 1*.

**Figure supplement 1.** Supplementary analysis of exponential phase *E. coli* sequencing data.

**Figure supplement 1—source code 1.** Related to *Figure 1—figure supplement 1*.

**Figure supplement 1—source data 1.** Related to *Figure 1—figure supplement 1*.

*C. crescentus* from exponential phase (*Figure 1B*; *Supplementary file 3*). Additionally, we compared our findings with other reported methods (*Figure 1B*; *Supplementary file 4*). The original PETRI-seq (*Blattman et al., 2020*) protocol, which does not include an rRNA depletion step, exhibited an mRNA detection rate of approximately 5%. The MicroSPLiT-seq (*Kuchina et al., 2021*) method, which utilizes poly A polymerase for mRNA enrichment, achieved a detection rate of 7%. Similarly, M3-seq (*Wang et al., 2023*) and BacDrop-seq (*Ma et al., 2023*), which employ RNaseH to digest rRNA post-DNA probe hybridization in cells, reported mRNA detection rates of 65% and 61%, respectively. MATQ-DASH (*Homberger et al., 2023*), which utilizes Cas9-mediated targeted rRNA depletion, yielded a detection rate of 30%. smRandom-seq utilizes a CRISPR-based rRNA depletion technique, reduced the rRNA proportion from 83% to 32%, increasing the mRNA proportion from 16% to 63% (*Xu et al., 2023*). BaSSSh-seq's employs a rational probe design for efficient rRNA depletion, though specific efficiency was not reported (*Korshoj and Kielian, 2024*). Among these, RiboD-PETRI demonstrated superior performance in mRNA detection while requiring the least sequencing depth. With equivalent sequencing depth, RiboD-PETRI demonstrates a significantly enhanced unique molecular identifier (UMI) counts detection rate compared to PETRI-seq alone (*Figure 1C*). This method recovered approximately 20,175 cells (92.6% recovery rate) with ≥15 UMIs per cell with a median UMI count of 42 per cell, which was significantly higher than PETRI-seq's recovery rate of 17.9% with a median UMI count of 20 per cell (*Figure 1—figure supplement 1A and B*), indicating the number of detected mRNA per cell increased prominently. Notably, this enhancement was achieved while maintaining mRNA profiles consistent with non-depleted samples (r=0.93; *Figure 1D*) and show a significant correlation with profiles from the traditional bulk RNA-seq method (r=0.84; *Figure 1E*).

We subsequently investigated the transcriptome coverage of RiboD-PETRI across different physiological states and bacterial species. For exponential phase *E. coli* cells, we sequenced a library with 60,000 cells, recovering approximately 30,004 cells (50% recovery), each with ≥15 UMIs (*Figure 2A*, *Figure 1—figure supplement 1C*). This analysis revealed 99.86% transcriptome-wide gene coverage across the cell population. The method achieved an average of 128.8 UMIs per single cell, with a median UMI count of 102 per cell. Further examination of high-quality cells showed varying levels of detection: the top 1000, 5000, and 10,000 cells exhibited median UMI counts of 462, 259, and 193, respectively (*Figure 2B*), and median gene detection of 362, 236, and 188, respectively (*Figure 2C*). These high-performing cells demonstrate the upper limits of the method's capabilities. For stationary phase *S. aureus* cells, we sequenced a library with 30,000 cells, recovering approximately 9982 cells (33.3% recovery), each with ≥15 UMIs (*Figure 2—figure supplement 1A*). Analysis showed 99.96% transcriptome-wide gene coverage across the cell population. At the single-cell level, we observed an average of 153.8 UMIs and a median of 142 UMIs. Top high-quality cells exhibited the following median UMI counts: 378 (top 1000 cells), 207 (top 5000 cells), and 167 (top 8000 cells) (*Figure 2—figure supplement 1B*). These cells also demonstrated median gene detection of 308, 194, and 158 genes, respectively (*Figure 2—figure supplement 1C*). For exponential phase *C. crescentus* cells, we sequenced a library with 30,000 cells, recovering approximately 13,897 cells (46.3% recovery), each with ≥15 UMIs (*Figure 2—figure supplement 1G*). Analysis showed 99.64% transcriptome-wide gene coverage across the cell population. At the single-cell level, we observed an average of 439.7 UMIs and a median of 182 UMIs. Top high-quality cells demonstrated the following median UMI counts: 2190 (top 1000 cells), 662 (top 5000 cells), and 225 (top 10,000 cells) (*Figure 2—figure supplement 1H*). These cells also exhibited median gene detection of 1262, 529, and 219 genes, respectively (*Figure 2—figure supplement 1I*). These results underscore RiboD-PETRI's ability to capture a wide

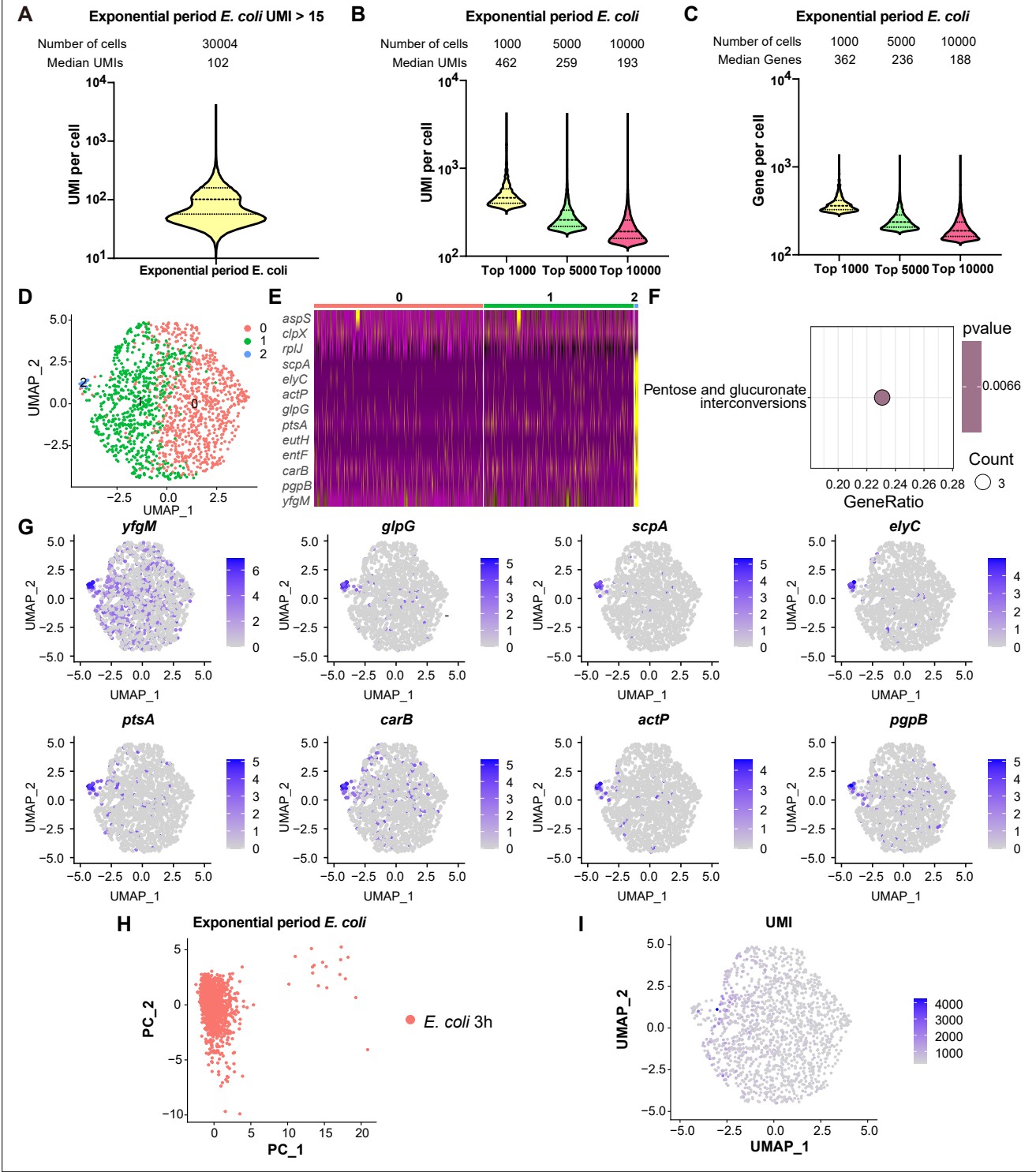

**Figure 2.** Comprehensive analysis of single-cell mRNA transcriptomic profiles in exponential phase *E. coli* using RiboD-PETRI. (**A**) The number of unique molecular identifiers (UMIs) detected per cell in recovered cells in exponential period *E. coli* (≥15 UMIs/cell). The cells are ranked from highest to lowest based on the number of detected UMIs, and cells with ≥15 UMIs are selected for plotting. The median number of UMIs is calculated for these selected cells. (**B**) Distribution of mRNA UMIs captured per cell in RiboD-PETRI data of exponential period *E. coli*, presented as violin plots showing the upper quartile, median, and lower quartile lines. The cells are ranked from highest to lowest based on the number of UMIs detected. Then, specific numbers of cells (indicated above the panel) are selected for plotting. The median number of UMIs is calculated for these selected cells. (**C**) The number of genes detected per cell in exponential period *E. coli*. The cells are ranked from highest to lowest based on the number of genes detected. Then, specific numbers of cells (indicated above the panel) are selected for plotting. The median number of genes is calculated for these selected cells.

*Figure 2 continued on next page*

*Figure 2 continued*

(**D**) Uniform Manifold Approximation and Projection (UMAP) visualization of *E. coli* bacteria during the exponential phase. Data were filtered for cells with UMIs between 200 and 5000, resulting in 1464 cells. Each dot represents a cell. (**E**) Heatmap illustrating the normalized gene expression levels of marker genes in different clusters of exponential period *E. coli*. Marker genes with relatively high expression levels are depicted in yellow, while lower expression levels are shown in purple. Each row represents a gene, and each column represents a cell. (**F**) Functional enrichment analysis of marker genes of exponential period *E. coli* in cluster 2. Marker genes were selected based on screening criteria of p-value <0.001 and $\log_2$ fold change (FC)>0.2. The color blocks in these figures represent the p-values of the data points. The color scale ranges from red to blue. Red colors indicate smaller p-values, suggesting higher statistical significance and more reliable results. Blue colors indicate larger p-values, suggesting lower statistical significance and less reliable results. Count is the number of genes enriched into this pathway. (**G**) Expression levels of marker genes in cluster 2 during the 3 hr exponential period of *E. coli* overlaid on the UMAP plot. Cells with high expression levels are depicted in blue. Marker genes were selected based on a p-value greater than 0.001 and a $\log_2$ FC greater than 3. (**H**) Principal component analysis (PCA) performed on screened data of exponential phase *E. coli*. The resulting scatterplots show heterogeneity among the populations, with each point representing a cell. (**I**) Distribution of UMIs on the UMAP results for exponential phase *E. coli*. UMAP results reveal heterogeneity among populations, with each point representing a cell and color shading indicating UMI counts (*Supplementary file 11*).

The online version of this article includes the following source data, source code, and figure supplement(s) for figure 2:

**Source code 1.** Source code for *Figure 2* and *Figure 2—figure supplement 2*.

**Source code 2.** Source code for *Figure 2—figure supplement 1*, *Figure 2—figure supplement 3* and *Figure 2—figure supplement 4*.

**Source data 1.** Related to *Figure 2*.

**Figure supplement 1.** Comprehensive single-cell transcriptomic analysis of *S. aureus* and *C. crescentus* using RiboD-PETRI.

**Figure supplement 1—source data 1.** Related to *Figure 2—figure supplement 1*.

**Figure supplement 2.** Profiling of marker genes in exponential phase *E. coli* culture by RiboD-PETRI.

**Figure supplement 2—source code 1.** Related to *Figure 2—figure supplement 2*.

**Figure supplement 3.** Marker genes identified in stationary phase *S. aureus* culture by RiboD-PETRI.

**Figure supplement 4.** Marker genes identified in exponential phase *C. crescentus* culture by RiboD-PETRI.

range of transcripts across varying cell qualities and species, providing a comprehensive view of gene expression at the single-cell level.

Our results affirm RiboD-PETRI's reliability in capturing the bacterial single-cell transcriptome, providing ample coverage and sensitivity for various species. To provide a thorough evaluation of our sequencing depth and library quality, we performed sequencing saturation analysis on our sequencing samples. The findings reveal that our sequencing saturation is greater than 90% (*Figure 1—figure supplement 1D–F*), indicating that our sequencing depth is sufficient to capture the diversity of most transcripts.

We further investigated its ability to consistently identify within-population heterogeneity across different bacterial species and growth conditions. In the exponential phase of *E. coli*, we recovered 1464 cells and identified three major subpopulations (*Figure 2D*), with 17 cells (1.2%) in a unique subpopulation characterized by pentose and glucuronate interconversions (*Figure 2E and F*) and the marker genes of cluster 2 included *yfgM*, *glpG*, *scpA*, *elyC*, *ptsA*, *carB*, *actP*, and *pgpB* (*Figure 2G*). For the expression levels of marker gene shown in *Figure 2E*, violin plots have been created to offer a more comprehensive view of the distribution across different cell populations (*Figure 2—figure supplement 2*). In stationary phase *S. aureus* cells, we recovered 9386 cells and found six major subpopulations (*Figure 2—figure supplement 1D*), with 437 cells (4.7%) in a distinct subpopulation named cluster 4. The marker genes of cluster 4 included KQ76-13335, KQ76-00740, and KQ76-11725 (*Figure 2—figure supplement 3*). In the stationary phase of *C. crescentus* cells, we recovered 5728 cells and identified four major subpopulations (*Figure 2—figure supplement 1J*), with 603 cells (10.5%) in a unique subpopulation named cluster 3. The marker genes of cluster 3 included CCNA-00259, CCNA-03402, CCNA-02361, and CCNA-03119 (*Figure 2—figure supplement 4*). These findings highlight RiboD-PETRI's consistent ability to unveil within-population heterogeneity across different cell physiology and bacterial species (*Figure 2H and I*, *Figure 2—figure supplement 1E, F, K, L*), crucial for understanding bacterial population complexity. While RiboD-PETRI consistently detects potential heterogeneity, further experimental validation would be required to confirm the biological significance of the observations.

We next focused on exploring biological heterogeneity of a biofilm at the early stage of development by utilizing the static biofilm system (*Merritt et al., 2011*). *E. coli* cells were cultured in microtiter

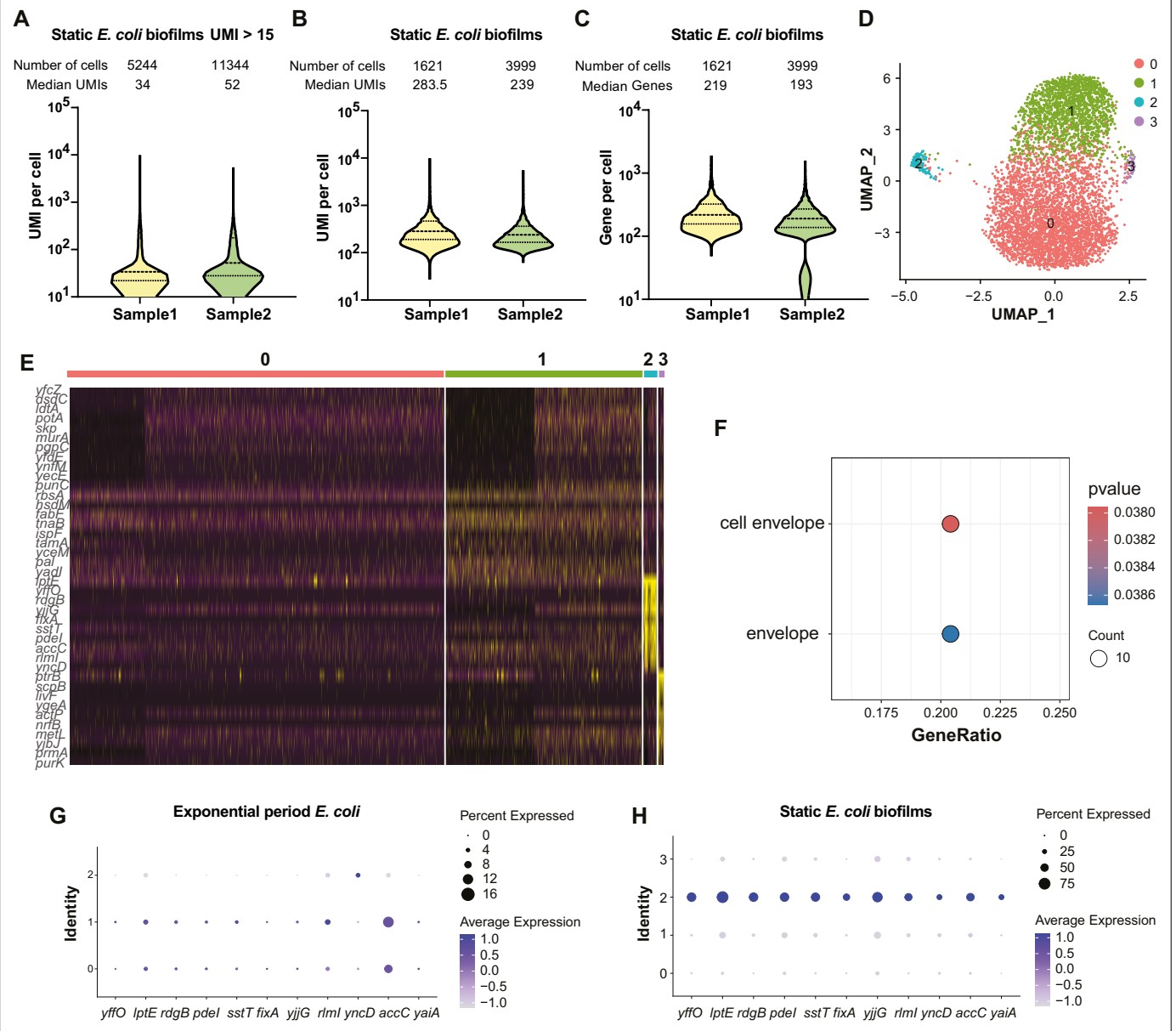

**Figure 3.** Single-cell transcriptomic analysis and characterization of static *E. coli* biofilm using RiboD-PETRI. (**A–F, H**) RiboD-PETRI data from static E. *coli* biofilm (*E. coli* 24 hr static culture) (***Supplementary files 12 and 13***). RiboD-PETRI data of static *E. coli* biofilm were screened for cells with unique molecular identifiers (UMIs) between 100 and 2000, resulting in 1621 and 3999 cells. (**A**) The number of UMIs detected per cell in recovered cells in Static *E. coli* biofilms (≥15 UMIs/cell). The cells are ranked from highest to lowest based on the number of detected UMIs, and cells with ≥15 UMIs are selected for plotting. (**B**) Distribution of mRNA UMIs captured per cell in RiboD-PETRI data of static *E. coli* biofilm. (**C**) The number of genes detected per cell in static *E. coli* biofilm. (**D**) UMAP visualization of static *E. coli* biofilm, revealing two small populations of heterogeneous cells in clusters 2 and 3. (**E**) Inferred expression levels of marker genes from static *E. coli* biofilm of *E. coli* across different clusters. (**F**) Enrichment pathways for marker genes of static *E. coli* biofilm data in cluster 2, selected based on screening criteria of p-value<0.001 and log$_2$ fold change (FC)>0.2. The color blocks in these figures represent the p-values of the data points. (G and H) Dot plot displaying scaled expression levels of marker genes in different clusters of *E. coli* in exponential phase (**G**) and *E. coli* in static *E. coli* biofilm (**H**). These genes were markers of static *E. coli* biofilms in cluster 2, identified with screening criteria of p-value<0.001 and log$_2$ FC>3. Dot size represents the percentage expression of the gene in the cluster, while color indicates the average expression level normalized from 0 to 1 across all clusters for each gene.

The online version of this article includes the following source data, source code, and figure supplement(s) for figure 3:

**Source code 1.** Source code for *Figure 3*, *Figure 3—figure supplement 1* and *Figure 3—figure supplement 2*.

**Source data 1.** Related to *Figure 3*.

**Figure supplement 1.** Evaluation of transcriptomic consistency and batch effect analysis in static biofilm *E. coli* samples.

*Figure 3 continued on next page*

*Figure 3 continued*

**Figure supplement 1—source data 1.** Related to *Figure 3—figure supplement 1*.

**Figure supplement 2.** Marker genes identified in static *E. coli* biofilms by RiboD-PETRI.

dishes overnight, adhered cells were fixed for RiboD-PETRI processing in duplicate experiments. For these two replicates, we sequenced libraries containing 20,000 and 40,000 cells, recovering 5244 and 11,344 cells, which corresponded to recovery rates of 26% and 28%, respectively. The correlations between detected reads and UMIs were found to be 0.87 and 0.90 for the two replicates, respectively (*Figure 3—figure supplement 1A*). The median UMI counts for the recovered cells were 34 and 52 (*Figure 3A*). After screening, the final datasets comprised 1621 and 3999 cells for the two replicates, respectively. While no significant batch effects were observed, we applied batch correction as a precautionary measure (*Figure 3—figure supplement 1B–D*). In replicate 1, each cell was sequenced with an average of 1563 reads, while in replicate 2, the average was 2034 reads (*Supplementary file 5*), yielding median UMI counts of 283.5 and 239 per cell, respectively (*Figure 3B*). For gene detection, the median counts were 219 and 193 genes per cell for the respective replicates (*Figure 3C*). Additionally, UMAP visualization was employed to illustrate the distribution of cellular UMI numbers, revealing heterogeneity among populations that was independent of UMI counts (*Figure 3—figure supplement 1E and F*). Unsupervised clustering analysis identified four major subpopulations in each replicate, with a consistently identified rare subpopulation (2.6%/2.1%) as cluster 2, driven by cell envelope genes (*Figure 3D–F*). Marker genes for this cluster included *yffO*, *lptE*, *rdgB*, *pdeI*, *sstT*, *fixA*, *yjjG*, *rlmI*, *accC*, and *yaiA* (*Figure 3G, H* and *Figure 3—figure supplement 2*).

PdeI, identified among marker genes, was predicted as a phosphodiesterase enzyme hydrolyzing c-di-GMP, a vital bacterial second messenger (*Yu et al., 2023*; *Li et al., 2023*; *Figure 4A and B*). However, our comprehensive structural analysis revealed a more complex and novel role for PdeI. While PdeI contains an intact EAL domain typically associated with c-di-GMP degradation, it also possesses a divergent GGDEF domain, generally linked to c-di-GMP synthesis (*Figure 4—figure supplement 1*). This dual-domain architecture suggested potential complex regulatory roles. To validate PdeI's function, we created a PdeI-BFP fusion construct under the native *pdeI* promoter, integrated with a ratiometric c-di-GMP sensing system (*Vrabioiu and Berg, 2022*) in *E. coli*. Confocal microscopy revealed PdeI as a membrane protein (*Figure 4C*). Single-cell level monitoring showed cell-to-cell variability in c-di-GMP levels and PdeI expression, with a positive correlation observed (*Figure 4D*), indicating PdeI upregulated c-di-GMP synthesis rather than degradation. This finding was confirmed by high-pressure liquid chromatography-tandem mass spectrometry (HPLC-MS/MS), which showed an approximately 11-fold increase in c-di-GMP concentration in the PdeI overexpression strain compared to the control strain (*Figure 4E*). These results align with previous studies showing that a point mutation (G412S) in PdeI's divergent GGDEF domain in a strain lacking PdeH, the major phosphodiesterase in *E. coli*, resulted in decreased c-di-GMP levels (*Reinders et al., 2016*). Our additional experiments with a PdeI(G412S)-BFP mutation strain showed constant c-di-GMP levels despite increasing BFP fluorescence, serving as a proxy for PdeI(G412S) expression levels (*Figure 4D*). These results, combined with the presence of a CHASE (cyclases/histidine kinase-associated sensory) domain in PdeI, strongly suggest that PdeI functions as a membrane-associated sensor that integrates environmental signals with c-di-GMP production under complex regulatory mechanisms. This discovery challenges the initial prediction of PdeI as solely a phosphodiesterase and highlights its novel role as a c-di-GMP synthetase, contributing significantly to our understanding of bacterial signaling pathways. It's worth noting that while the other marker genes in this cluster are co-expressed, our analysis indicates that they do not have a significant impact on biofilm formation or a direct relationship with c-di-GMP or PdeI.

Confocal laser scanning microscopy provided further insights into the spatial distribution of PdeI-positive cells within the biofilm structure. In the PdeI-BFP fusion strain, PdeI-BFP-positive cells, characterized by elevated c-di-GMP levels, were predominantly located at the bottom of the static biofilm (*Figure 4F*). This localization corresponds to the region of cell-surface attachment, aligning with our hypothesis that PdeI functions as a membrane-associated sensor integrating environmental signals with c-di-GMP production through complex regulatory mechanisms (*Lacanna et al., 2016*). In contrast, in the control strain where BFP was expressed alone under arabinose-induced promoter, BFP-positive cells were observed to be distributed throughout the entire biofilm community (*Figure 4G*). This distinct spatial distribution pattern between the PdeI-BFP fusion and the BFP-only control strains

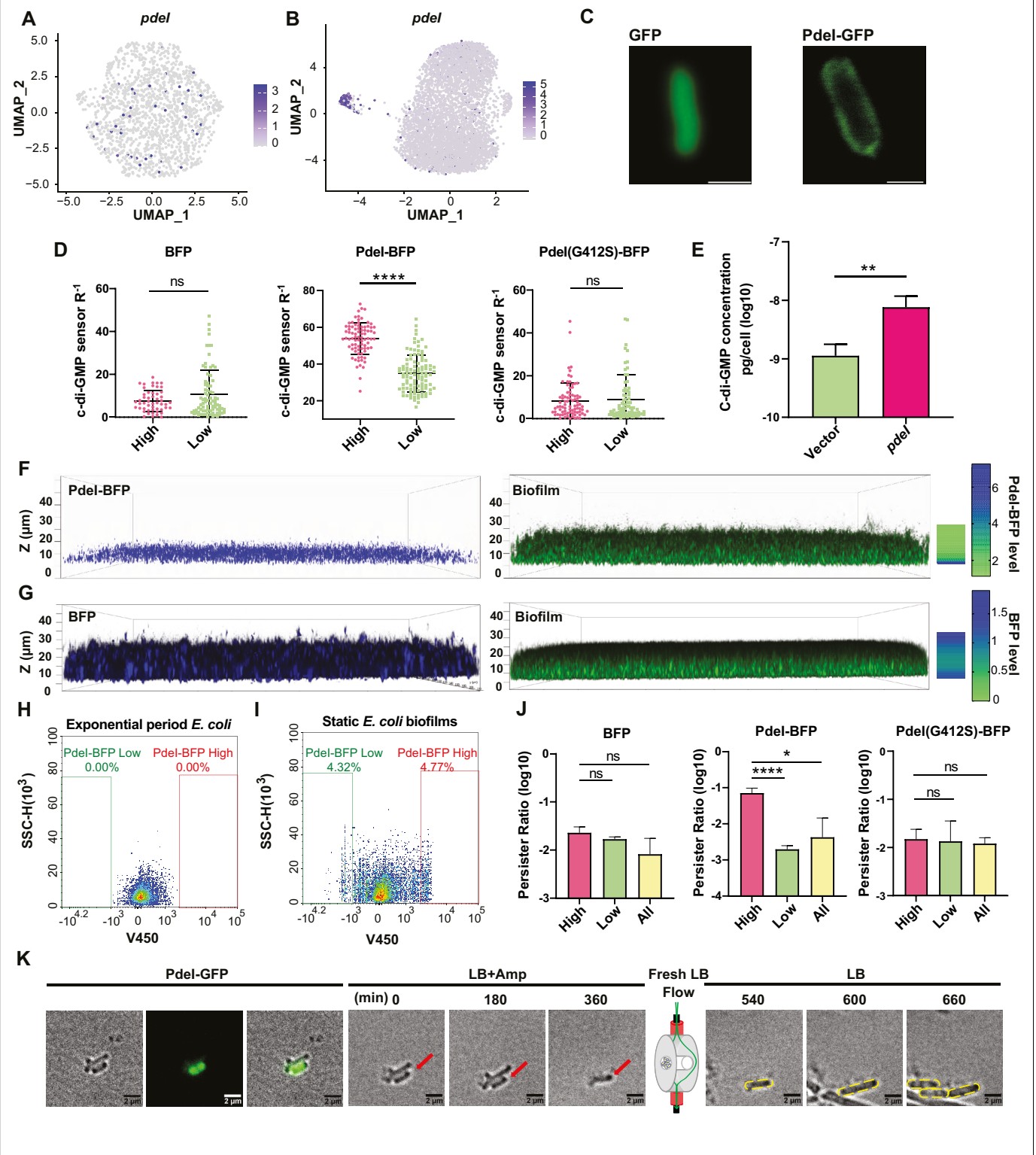

**Figure 4.** Functional investigation of marker gene *pdeI* in static *E. coli* biofilm. (**A, B**) Uniform Manifold Approximation and Projection (UMAP) plots showing the distribution of *pdeI* in single-cell data of exponential period *E. coli* (**A**) and static *E. coli* biofilm (**B**). Each dot represents a cell colored by normalized expression levels of genes. (**C**) Subcellular localization of PdeI-GFP and GFP. Scale bar, 1 μm. (**D**) c-di-GMP levels ($R^{-1}$ score) in *E. coli* cells with different BFP, PdeI-BFP, PdeI(G412S)-BFP expression levels (low or high), under the control of the *pdeI* native promoter, in static *E. coli* biofilm. c-di-GMP levels are measured using the c-di-GMP sensor system integrated into *E. coli* cells. $R^{-1}$ score was determined using the fluorescent intensity of mVenusNB and mScarlet-I in the system. The fluorescent intensity is measured by flow cytometry (n>50). (**E**) Determination of cellular concentrations

*Figure 4 continued on next page*

*Figure 4 continued*

of c-di-GMP by high-pressure liquid chromatography-tandem mass spectrometry (HPLC-MS/MS) in cells overexpressing PdeI under the control of arabinose promoter, with 0.002% arabinose induction for 2 hr (n=3). (**F, G**) Localization of PdeI-high cells in the biofilm matrix. Cells expressing PdeI-BFP under the control of the *pdeI* native promoter were grown in a glass-bottom cell culture dish and stained with SYTO 24 for bacterial DNA. Cells expressing BFP under the control of arabinose promoter, with 0.00001% arabinose induction for 24 hr in a glass-bottom cell culture dish and stained with SYTO 24 for bacterial DNA. (**H, I**) Heterogeneous expression of PdeI in single-cell data of exponential period *E. coli* (**H**) and *E. coli* in static *E. coli* biofilm (*E. coli* 24 hr static culture) (**I**). Biofilm cells with high or low expression levels of PdeI-BFP were sorted by flow cytometry. (**J**) Persister counting assay using 150 μg/ml ampicillin on cells with high or low expression levels of BFP, PdeI-BFP, and PdeI(G412S)-BFP from static *E. coli* biofilm, sorted by flow cytometry (n=3). These strains were under the control of the *pdeI* native promoter. (**K**) Time-lapse images of the persister assay observed under a microscope. Static biofilm cells of the PdeI-GFP strain were spotted on a gel pad and treated with 150 μg/ml ampicillin in Luria broth (LB). Images were captured over 6 hr at 37°C, followed by the replacement of fresh LB to allow persister cell resuscitation. Scale bar, 2 μm. Error bars represent standard deviations of biological replicates. Significance was ascertained by unpaired Student's t-test. Statistical significance is denoted as *p<0.05, **p<0.01, ***p<0.001, and ****p<0.0001.

The online version of this article includes the following source data and figure supplement(s) for figure 4:

**Source data 1.** Related to *Figure 4*.

**Figure supplement 1.** Schematic chart for the structure of *E. coli* PdeI.

provides compelling evidence for PdeI's specific role in biofilm formation, particularly at the biofilm-surface interface. The concentration of PdeI-positive cells at the bottom of the biofilm suggests that PdeI may be especially crucial in the initial stages of biofilm formation, potentially responding to surface-associated cues to modulate c-di-GMP levels and promote attachment. The uniform distribution of BFP in the control strains suggests that the localization of PdeI-BFP is not affected by BFP labeling. These observations further underscore the complex and nuanced role of PdeI in bacterial signaling and biofilm development, highlighting the importance of considering cell-to-cell heterogeneity in understanding the function of regulatory proteins in microbial communities.

To investigate the association of the PdeI-high cluster with bacterial drug tolerance in the early stages of biofilm development, we isolated PdeI-high cells using flow cytometry (*Figure 4H, I*) and subjected them to an ampicillin antibiotic killing assay to determine their persister frequency. Our results revealed that the PdeI-high population produced a significantly higher ratio of persister cells (~7.3%) compared to the whole biofilm population (~0.6%). Notably, cells expressing high levels of BFP alone or PdeI(G412S)-BFP showed no increase in persister ratios (*Figure 4J*). This finding suggests that the increased persistence is specifically linked to PdeI activity. Time-lapse imaging during the antibiotic killing process consistently demonstrated that persisters primarily originated from PdeI-GFP-positive cells (*Figure 4K* and *Video 1*). These PdeI-GFP-positive cells, displaying characteristics of dormancy, survived ampicillin treatment for 6 hr without visible growth or division. Upon antibiotic removal and replacement with fresh growth medium, the PdeI-GFP-positive persister cells resumed activity, elongating, dividing, and forming new microcolonies (*Figure 4K* and *Video 1*). This dynamic behavior provides direct visual evidence of the persister phenotype associated with PdeI-high cells. These findings strongly suggest that c-di-GMP, a molecule whose intracellular levels are upregulated by PdeI, plays a significant role in generating a persister subpopulation during the early stages of biofilm development. The mechanism by which elevated c-di-GMP levels contribute to antibiotic tolerance may involve modulation of cellular metabolism or activation of stress response pathways, leading to a state of dormancy that enables survival under antibiotic stress. This discovery not only enhances our understanding of the link between biofilm formation and antibiotic tolerance but also identifies PdeI as a potential target for strategies aimed at combating persistent bacterial infections.

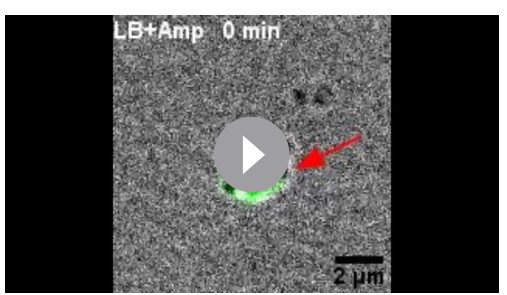

**Video 1.** Time-lapse images of the persister assay using cells with different PdeI-BFP.
https://elifesciences.org/articles/97543/figures#video1

## Discussion

In this study, we introduce RiboD-PETRI, an enhanced bacterial scRNA-seq method that offers a cost-effective (*Supplementary file 6*), equipment-free, and high-throughput solution.

By incorporating a probe hybridization-based rRNA-derived cDNA depletion protocol, our approach efficiently removes rRNA reads and significantly improves mRNA detection rates, enabling a more comprehensive exploration of within-population heterogeneity. At $0.0049 per cell, RiboD-PETRI is substantially more economical than the original PETRI-seq ($0.056 per cell), making it an attractive option for budget-conscious researchers. The method demonstrates improved mRNA detection, versatility across various bacterial species and growth conditions, preservation of transcriptome profiles consistent with non-depleted samples (r=0.93) and traditional bulk RNA-seq methods (r=0.84), high transcriptome coverage (>99%), and robust single-cell resolution with median UMI counts ranging from 102 to 182 per cell across different species and conditions.

The application of RiboD-PETRI to investigate biofilm heterogeneity exemplifies its potential for exploring complex biological systems. Our analysis of early-stage biofilm development uncovered a rare subpopulation (2.1–2.6%) characterized by cell envelope genes, including the previously uncharacterized gene *pdeI*. Further investigation revealed PdeI's novel role as a c-di-GMP synthetase rather than a phosphodiesterase, challenging initial predictions and contributing significantly to our understanding of bacterial signaling pathways. Moreover, we demonstrated that PdeI-high cells exhibit increased antibiotic tolerance, with a significantly higher proportion of persister cells compared to the general biofilm population. This finding establishes a link between elevated c-di-GMP levels, regulated by PdeI, and the generation of antibiotic-tolerant subpopulations during early biofilm development. The spatial distribution of PdeI-positive cells at the bottom of the static biofilm, corresponding to the cell-surface attachment region, supports our hypothesis that PdeI functions as a membrane-associated sensor integrating environmental signals with c-di-GMP production. This localization pattern suggests PdeI's crucial role in the initial stages of biofilm formation, potentially responding to surface-associated cues to modulate c-di-GMP levels and promote attachment. While other marker genes in this cluster are co-expressed, our analysis indicates they do not significantly impact biofilm formation or directly relate to c-di-GMP or PdeI.

In conclusion, RiboD-PETRI represents a significant advancement in bacterial scRNA-seq methodology. Its ability to uncover hidden variations within bacterial populations, as demonstrated in our biofilm analysis, underscores its potential impact on advancing our understanding of microbial behavior and population dynamics. By providing a cost-effective and efficient tool for exploring bacterial heterogeneity, RiboD-PETRI opens new avenues for research in microbiology, potentially leading to novel insights into antibiotic resistance, biofilm formation, and other critical areas of bacterial biology.

# Materials and methods

## Key resources table

| Reagent type (species) or resource | Designation | Source or reference | Identifiers | Additional information |
|---|---|---|---|---|
| Strain, strain background (*Escherichia coli*) | MG1655 | Yale Genetic Stock Center | CGSC#6300 | |
| Strain, strain background (*Caulobacter crescentus*) | NA1000 | Shenzhen Institutes of Advanced Technology, Chinese Academy of Sciences | NCBI accession number CP001340 | |
| Strain, strain background (*Staphylococcus aureus*) | ATCC 25923 | ATCC | ATCC 25923 | |
| Strain, strain background (*Escherichia coli*) | MG1655 pBAD::*gfp* | This paper | | Figure legends and Materials and methods section |
| Strain, strain background (*Escherichia coli*) | MG1655 p(*pdeI* promoter)::*pdeI-gfp* | This paper | | Figure legends and Materials and methods section |
| Strain, strain background (*Escherichia coli*) | MG1655 p(*pdeI* promoter)::*pdeI-bfp* | This paper | | Figure legends and Materials and methods section |
| Strain, strain background (*Escherichia coli*) | MG1655 Δ*ara* pBAD::*pdeI* | This paper | | Figure legends and Materials and methods section |
| Strain, strain background (*Escherichia coli*) | MG1655 Δ*ara* pBAD::vector | This paper | | Figure legends and Materials and methods section |
| Strain, strain background (*Escherichia coli*) | MG1655 p(*pdeI* promoter)::*bfp* | This paper | | Figure legends and Materials and methods section |

*Continued on next page*

*Continued*

| Reagent type (species) or resource | Designation | Source or reference | Identifiers | Additional information |
|---|---|---|---|---|
| Strain, strain background (*Escherichia coli*) | MG1655 p(*pdel* promoter)::*pdel*(G412S)-*bfp* | This paper | | Figure legends and Materials and methods section |
| Strain, strain background (*Escherichia coli*) | MG1655 p(*pdel* promoter)::*bfp* p15A::c-di-GMP-sensor | This paper | | Figure legends and Materials and methods section |
| Strain, strain background (*Escherichia coli*) | MG1655 p(*pdel* promoter)::*pdel-bfp* p15A::c-di-GMP-sensor | This paper | | Figure legends and Materials and methods section |
| Strain, strain background (*Escherichia coli*) | MG1655 p(*pdel* promoter)::*pdel*(G412S)-*bfp* p15A::c-di-GMP-sensor | This paper | | Figure legends and Materials and methods section |
| Strain, strain background (*Escherichia coli*) | MG1655 Δ*ara* pBAD::*bfp* | This paper | | Figure legends and Materials and methods section |
| Recombinant DNA reagent | p15A::c-di-GMP-sensor | This paper | | p15A ori |
| Recombinant DNA reagent | pBAD::vector | This paper | | Arabinose-induction |
| Recombinant DNA reagent | pBAD::*gfp* | This paper | | Arabinose-induction |
| Recombinant DNA reagent | pBAD::*bfp* | This paper | | Arabinose-induction |
| Recombinant DNA reagent | p(*pdel* promoter)::*bfp* | This paper | | *pdel* native promoter induction |
| Recombinant DNA reagent | p(*pdel* promoter)::*pdel-bfp* | This paper | | *pdel* native promoter induction |
| Recombinant DNA reagent | p(*pdel* promoter)::*pdel*(G412S)-*bfp* | This paper | | *pdel* native promoter induction |
| Recombinant DNA reagent | p(*pdel* promoter)::*pdel-gfp* | This paper | | *pdel* native promoter induction |
| Recombinant DNA reagent | p(*pdel* promoter)::*pdel* | This paper | | *pdel* native promoter induction |
| Recombinant DNA reagent | pBAD::*pdel-gfp* | This paper | | Arabinose-induction |
| Sequence-based reagent | P-pdel-F | This paper | PCR primers | AATTGTCTGATTCGTTACC AACTGACCGTACTGGCGTTC |
| Sequence-based reagent | P-pdel-R | This paper | PCR primers | TTGCTGCTGCCTCGGCTTC TAGCTCTTTTACTAATTTTCC ACTTTTATCCCAGG |
| Sequence-based reagent | pdel-F | This paper | PCR primers | GGCTAACAGGAGGAATTAA CCATGCTGAGTTTATACGA AAAGATAAAGATAAG |
| Sequence-based reagent | pdel-R | This paper | PCR primers | GCTGGAGACCGTTTAAACT CACTACTCTTTTACTAATTT TCCACTTTTATCCC |
| Sequence-based reagent | pBAD-R | This paper | PCR primers | TTGGTAACGAATCAGACAATTGAC |
| Sequence-based reagent | pBAD-F | This paper | PCR primers | TGAGTTTAAACGGTCTCCAGC |
| Sequence-based reagent | pBAD-R2 | This paper | PCR primers | GGTTAATTCCTCCTGTTAGCCC |
| Sequence-based reagent | Bfp-F | This paper | PCR primers | CGAGGCAGCAGCAAAGGCCC TAGAAGGTGGATCCGGCGGTTCTAG |
| Sequence-based reagent | Gfp-F | This paper | PCR primers | CTAGAAGCCGAGGCAGCAGC AAAGGCCCTAGAAATGAGTAAA GGAGAAGAACTTTTCAC |
| Sequence-based reagent | G412S-F | This paper | PCR primers | GAAGCGGTGTTTAGTGTTGATG |
| Sequence-based reagent | G412S-R | This paper | PCR primers | CATCAACACTAAACACCGCTTC |
| Sequence-based reagent | P-bfp-R | This paper | PCR primers | GTTAATACATTTAACAAAA TAACTATCTGA |
| Sequence-based reagent | P-bfp-F | This paper | PCR primers | ATAGTTATTTTGTTAAATGTA TTAACGGTGGATCCGGCGGTTCT |
| Sequence-based reagent | UP-F | This paper | PCR primers | CATGAATTCTGGCGACGATTTCG |
| Sequence-based reagent | UP-R | This paper | PCR primers | GTTAATACATTTAACAAAA TAACTATCTGA |

*Continued on next page*

*Continued*

| Reagent type (species) or resource | Designation | Source or reference | Identifiers | Additional information |
|---|---|---|---|---|
| Sequence-based reagent | ccdB-F | This paper | PCR primers | CACAGCGTTCAGATAGTTATTT TGTTAAATGTATTAACTCTAG AGCGACGCCAGACG |
| Sequence-based reagent | ccdB-R | This paper | PCR primers | CTGTAAGTACGAACTTATTGAT TCTGGACATACGTAAATTAC GCCCCGCCCTGCCAC |
| Sequence-based reagent | Down-F | This paper | PCR primers | TTTACGTATGTCCAGAATCA ATAAGTTCGTACTTAC |
| Sequence-based reagent | Down-R | This paper | PCR primers | ATCTTCGTCAAAGGATTTTCTGCCC |
| Sequence-based reagent | UP2-R | This paper | PCR primers | ATCTTTTCGTATAAACTCAG CATGTTAATACATTTAAC AAAATAACTATCTGAA |
| Sequence-based reagent | pdel-G412S-F | This paper | PCR primers | ATGCTGAGTTTATACGAAAAGATAAAGAT |
| Sequence-based reagent | pdel-G412S-R | This paper | PCR primers | CTTATTGATTCTGGACATACGT AAACTACTCTTTACTAATTTTCCACT |
| Sequence-based reagent | Down2-F | This paper | PCR primers | TTTACGTATGTCCAGAATCA ATAAGTTCGTACTTAC |
| Commercial assay or kit | KAPA HIFI hotStart ReadyMix PCR Kits | KAPA | Cat#2602 | |
| Commercial assay or kit | VAHTS Universal DNA Library Prep Kit | Vazyme | Cat#NR603 | |
| Commercial assay or kit | Bacteria RNA Extraction Kit | Vazyme | Cat#R403-01 | |
| Commercial assay or kit | Ribo-off rRNA Depletion Kit (Bacteria) | Vazyme | Cat#N407 | |
| Commercial assay or kit | 2× MultiF Seamless Assembly Mix | ABclonal | Cat#RK21020 | |
| Commercial assay or kit | VAHTS Universal DNA Library Prep Kit for Illumina V3 | Vazyme | Cat#ND607 | |
| Commercial assay or kit | ABScript III RT Master Mix for qPCR with gDNA Remover | ABclonal | Cat#RK20429 | |
| Commercial assay or kit | SUPERase-In RNase Inhibitor | Invitrogen | Cat#AM2696 | |
| Chemical compound, drug | Streptavidin Magnetic Beads | Thermo Fisher | Cat#88816 | |
| Chemical compound, drug | Syto 24 dye | Invitrogen | Cat#S7559 | |
| Chemical compound, drug | Arabinose | Sigma | Cat#V900920 | |
| Chemical compound, drug | Ampicillin | Sangon Biotech | Cat#A610028 | |
| Chemical compound, drug | Chloramphenicol | Sangon Biotech | Cat#A600118 | |
| Chemical compound, drug | Kanamycin | Sangon Biotech | Cat#A600286 | |
| Software, algorithm | Fiji | GitHub | https://fiji.sc/; RRID:SCR_002285 | |
| Software, algorithm | FlowJo | Treestar, Inc | https://www.flowjo.com/ | |

## Resource availability

Further information and requests for resources and reagents should be directed to and will be fulfilled by the lead contact, Yingying Pu (yingyingpu@whu.edu.cn).

## Materials availability

Plasmids generated in this study are available from the lead contact upon request.

## Bacterial strains and growth conditions

The bacterial strains used in this study included *E. coli* strains MG1655, *C. crescentus* NA1000, and *S. aureus* strain ATCC 25923. *E. coli* cultures were grown in Luria broth (LB) medium. For the biofilm setup, bacterial cultures were grown overnight. The next day, we diluted the culture 1:100 in a Petri dish. We added 2 ml of LB medium to the dish. If the bacteria contain a plasmid, the appropriate antibiotic needs to be added to LB. The Petri dish was then incubated statically in a growth chamber for 24 hr. After incubation, we performed imaging directly under the microscope. The Petri dishes

used were glass-bottom dishes from Biosharp (catalog number BS-20-GJM), allowing for direct microscopic imaging without the need for cover slips or slides. This setup allowed us to grow and image the biofilms in situ, providing a more accurate representation of their natural structure and composition. *C. crescentus* strain NA1000 was grown in peptone yeast extract (PYE) medium. And *S. aureus* strain ATCC 25923 was grown in Mueller-Hinton Broth (MHB) medium. All bacterial strains were routinely grown at 37°C and 220 rpm. To maintain plasmids, when necessary, media were supplemented with chloramphenicol (25 µg/ml) or kanamycin sulfate (50 µg/ml). For arabinose-induction system expression experiments, 0.002% or 0.00002% arabinose was supplemented in the medium.

### Strains construction
The construction of recombinant plasmids was performed using the 2× MultiF Seamless Assembly Mix (ABclonal, RK21020). For the detection of c-di-GMP levels using c-di-GMP sensor and the detection of persister, the PdeI gene, *pdeI*, along with its native promoter (250 bp), was fused with either *gfp* or *bfp* and cloned into the pBAD backbone. The original promoter region of the pBAD vector was removed to avoid any potential interference. This construction allows the expression of the BFP, PdeI-BFP, and PdeI(G412S)-BFP fusion proteins to be driven by *pdeI*'s native promoter, thus maintaining its physiological control mechanisms. And the BFP coding sequence was fused to the *pdeI* gene to create the PdeI-BFP fusion construct. Besides, for membrane localization and localization in biofilm community, the *pdeI-gfp* and *pdeI-bfp* with native promoter of *pdeI* were cloned into the pBAD backbone, and the original promoter region of the pBAD vector was removed. For the control group, *bfp* and *gfp* was cloned into the pBAD backbone under the control of arabinose-induction system. For HPLC-MS/MS analysis, *pdeI* and empty vector were cloned into the pBAD backbone, induced by arabinose. GFP and BFP were used in different experiments. GFP was used for imaging and time-lapse imaging to observe persister cell growth. BFP was used for cell sorting and detecting the proportion of persister cells. For the c-di-GMP sensor (Addgene: #182291), the plasmid origin was replaced with the p15A ori.

## RiboD-PETRI
### Cell preparation
*E. coli* MG1655 cells were cultured overnight and subsequently diluted at a ratio of 1:100 into fresh LB medium and grown statically for 24 hr at 37°C. For 3 hr exponential period *E. coli* sample, *E. coli* MG1655 cells were grown overnight and then diluted 1:100 into fresh LB medium and grown for 3 hr at 37°C and 220 rpm. *C. crescentus* strain NA1000 cells were grown overnight and then diluted 1:100 into fresh MHB medium and grown for 9 hr at 37°C and 220 rpm. And *S. aureus* strain ATCC 25923 cells were grown overnight and then diluted 1:100 into fresh PYE medium and grown for 3 hr at 37°C and 220 rpm. All the culture was vigorously shaken using a vortex, and the cells were then centrifuged at 5000×*g* for 2 min at 4°C. The pellet was resuspended in 2 ml of ice-cold 4% formaldehyde (F8775, MilliporeSigma, diluted into PBS). These suspensions were rotated at 4°C for 16 hr.

### Cell permeabilization
1 ml of fixed cells were centrifuged at 5000×*g* for 5 min at 4°C, then resuspended in 1 ml washing buffer (100 mM Tris-HCl pH 7.0, 0.02 U/µl SUPERase-In RNase Inhibitor, AM2696, Invitrogen). After another centrifugation at 5000×*g* for 5 min at 4°C, the supernatant was removed. The pellet was then resuspended in 250 µl permeabilization buffer (0.04% Tween-20 in PBS-RI, PBS with 0.01 U/µl SUPERase-In RNase Inhibitor) and incubated on ice for 3 min. 1 ml cold PBS-RI was added, and the cells were centrifuged at 5000×*g* for 5 min at 4°C. The pellet was resuspended in 250 µl Lysozyme Mix (250 µg/ml Lysozyme or 5 µg/ml Lysostaphin for *S. aureus* in TEL-RI buffer, comprising 100 mM Tris pH 8.0 [AM9856, Invitrogen], 50 mM EDTA [AM9261, Invitrogen], and 0.1 U/µl SUPERase In RNase Inhibitor). The samples were incubated at 37°C and mixed gently every minute. Then 1 ml cold PBS-RI was added immediately, and cells were centrifuged at 5000×*g* for 5 min at 4°C. The cells underwent another wash with 1 ml cold PBS-RI. Subsequently, cells were resuspended in 40 µl DNaseI-RI buffer (4.4 µl 10× reaction buffer, 0.2 µl SUPERase In RNase inhibitor, 35.4 µl H₂O), followed by addition of 4 µl DNaseI (AMPD1, MilliporeSigma). The samples were incubated for 30 min at room temperature and mixed gently every 5 min. 4 µl Stop Solution was added, and the samples were incubated for 10 min at 50°C with gentle mixing every minute. Following centrifugation at 5000×*g* for 10 min at

4°C, cells were washed twice with 0.5 ml cold PBS-RI. Finally, cells were resuspended in 200 µl cold PBS-RI, and their count and integrity were assessed using the ACEA NovoCyte flow cytometer with a 100× oil immersion lens.

## Primer preparation

For the first round of reverse transcription reaction, round 2 and round 3 ligation reactions, all primers design and preparation as previously described (*Blattman et al., 2020*). All primers were purchased from Sangon Biotech (*Supplementary file 1*). For ligation primers preparation, mixtures were prepared as follows: 31.1 µl each R2 primer (100 µM), 28.5 µl SB83 (100 µM), and 21.4 µl H₂O were splitted to 2.24 µl for one sample. Mixtures containing 63.2 µl each R3 primer (70 µM) and 58 µl SB8 (70 µM) were splitted to 3.49 µl for one sample. Before use, ligation primers were incubated as follows: 95°C for 3 min, then decreasing the temperature to 20°C at a ramp speed of −0.1 °C/s, 37°C for 30 min. For blocking mix preparation, 50 µl primer SB84 (400 µM) and 80 µl primer SB81 (400 µM) were incubated as follows: 94°C for 3 min, then decreasing the temperature to 25°C at a ramp speed of −0.1 °C/s, 4°C for keeping. Round 2 blocking primers were mixed as follows: 37.5 µl 400 µM SB84, 37.5 µl 400 µM SB85, 25 µl 10× T4 ligase buffer, 150 µl H₂O. Round 3 blocking primers were mixed as follows: 72 µl 400 µM SB81, 72 µl 400 µM SB82, 120 µl 10× T4 ligase buffer, 336 µl H₂O, 600 µL 0.5 M EDTA.

## Round 1 RT reaction

About $3×10^7$ cells were introduced into an RT reaction mix composed of 240 µl 5× RT buffer, 24 µl dNTPs (N0447L, NEB), 12 µl SUPERase In RNase Inhibitor, and 24 µl Maxima H Minus Reverse Transcriptase (EP0753, Thermo Fisher Scientific), 132 µl PEG8000 (50%). Nuclease-free water was added to achieve a total reaction volume of 960 µl, and the mixture was thoroughly mixed by vortexing. Subsequently, 8 µl of the reaction mixture was dispensed into each well of a 96-well plate, where 2 µl of each RT primer had been added previously. The sealed 96-well plate was inverted repeatedly for thorough mixing, followed by a brief spin. The plate was then incubated as follows: 50°C for 10 min, 8°C for 12 s, 15°C for 45 s, 20°C for 45 s, 30°C for 30 s, 42°C for 6 min, 50°C for 16 min, and finally held at 4°C. After the RT process, all 96 reactions were pooled into one tube. 75 µl of 0.5% Tween-20 was added, and the reactions were incubated on ice for 3 min. Cells were centrifuged at 7000×*g* for 10 min at 4°C and then resuspended in 0.4 ml PBS-RI. Thirty-two microliters of 0.5% Tween-20 was added, and the cells underwent centrifugation at 7000×*g* for 10 min at 4°C.

## Round 2 ligation reaction

Cells were resuspended in 500 µl 1× T4 ligase buffer, followed by the addition of 107.5 µl PEG8000, 37.5 µl 10× T4 ligase buffer, 16.7 µl SUPERase In RNase Inhibitor, 5.6 µl BSA, and 27.9 µl T4 ligase (M0202L, NEB). The reaction solution was thoroughly mixed by vortexing. Subsequently, 5.76 µl of the reaction mixture was dispensed into each well of a 96-well plate, where 2.24 µl of each round 2 ligation primer had been added previously. The sealed 96-well plate was inverted repeatedly for thorough mixing and then subjected to a short spin. The plate was incubated at 37°C for 45 min. Following this, 2 µl of round 2 blocking mix was added to each well and incubated at 37°C for an additional 45 min. All 96 reactions were pooled into one tube after incubation.

## Round 3 ligation reaction

A mixture comprising 89 µl H₂O, 26 µl PEG8000, 46 µl 10× T4 ligase buffer, and 12.65 µl T4 ligase was prepared and thoroughly mixed by vortexing. Subsequently, 8.51 µl of the reaction mixture was dispensed into each well of a 96-well plate, where 3.49 µl of each round 3 ligation primer had been added previously. The sealed 96-well plate was inverted repeatedly for thorough mixing and then subjected to a brief spin. The plate was incubated at 37°C for 45 min. Following this, 10 µl of round 3 blocking mix was added to each well and incubated at 37°C for an additional 45 min. All 96 reactions were combined into one tube after incubation.

## Cells lysis

42 µl of 0.5% Tween-20 was added, and cells were centrifuged at 7000×*g* for 10 min at 4°C. The cells underwent two washes using 200 µl TEL-RI containing 0.01% Tween-20, each time centrifuged at

7,000×g for 10 min at 4°C. Subsequently, cells were resuspended in 30 µl TEL-RI buffer. Cell counting and integrity checks were performed using the ACEA NovoCyte flow cytometer with a 100× oil immersion lens. A moderate amount of cells was then added to the lysis buffer (50 mM Tris pH 8.0, 25 mM EDTA, 200 mM NaCl, 0.5% Triton X-100), and 5 µl of proteinase K (AM2548, Invitrogen) was introduced. Samples were incubated at 55°C for 60 min and gently mixed every minute.

## Library construction

To facilitate template switching, lysates were purified with VAHTS DNA Clean Beads (N411, Vazyme) at a ratio of 2.0×, and cDNA was eluted in 12 µl of water. The purified cDNA was then combined with 4 µl of 5× RT buffer, 1 µl of dNTPs (N0447L, NEB), 0.5 µl of SUPERase In RNase Inhibitor, 0.5 µl of Maxima H Minus Reverse Transcriptase, and 2 µl of the TSO (*Picelli et al., 2013*) primer (100 mM, *Supplementary file 1*). This reaction solution underwent incubation as follows: 25°C for 30 min, 42°C for 90 min, 85°C for 5 min, and then held at 4°C. Subsequently, 1 µl of RNaseH was added, and the reaction solution was incubated at 37°C for 30 min. The cDNA was purified once again with VAHTS DNA Clean Beads at a ratio of 2.0× and eluted in 13 µl of $H_2O$. The integrity of the cDNA was assessed using primers TSO-2 and R1 or R2 or R3 by qPCR (*Supplementary file 1*).

## Ribosomal RNA-derived cDNA depletion

We developed a set of cDNA probe primers to selectively deplete r-cDNA (*Supplementary file 1*). These probe primers possess the ability to specifically hybridize with r-cDNA and also hybridize with a biotin-labeled universal primer. In the reaction, 5 µl of r-cDNA probe primers (10 µM), 2.5 µl of 10× hybridization buffer (Tris-HCl pH 8.0 100 mM, NaCl 500 mM, EDTA pH 8.0 10 mM), and 5 µl of biotin primer (10 µM) were added to 12.5 µl of purified cDNA. The reaction solution underwent incubation as follows: 95°C for 2 min, followed by a temperature decrease to 20°C at a ramp speed of −0.1 °C/s, and then held at 37°C for 30 min. Subsequently, 20 µl of Streptavidin magnetic beads (BEAVER, 22307) was washed twice using 1 ml of 1× B&W buffer (Tris-HCl pH 7.5 10 mM, EDTA 1 mM, NaCl 1 M, Tween-20 0.05%) and resuspended in 25 µl of 2× B&W buffer. Twenty-five microliters of washed Streptavidin magnetic beads were added to 25 µl of annealed cDNA. The reaction solution was incubated at room temperature for 30 min with gentle mixing per minute. Following this, the reaction solution tube was placed into a magnetic stand to collect the supernatant. The cDNA depleted of r-cDNA was purified using VAHTS DNA Clean Beads at a ratio of 2.0× and eluted in 12.5 µl of $H_2O$. The depletion of r-cDNA could be repeated using the above protocol, and ultimately, the cDNA was eluted in 20 µl of $H_2O$. We designed separate probe sets for *E. coli*, *C. crescentus*, and *S. aureus*. Each set was specifically constructed to be reverse complementary to the r-cDNA sequences of its respective bacterial species. This species-specific approach ensures high efficiency and specificity in rRNA depletion for each organism.

## Library amplification and sequencing

To the 20 µl cDNA solution, the following components were added: 2.4 µl R3 primer (10 mM, *Supplementary file 1*), 2.4 µl TSO-2 primer (10 mM, *Supplementary file 1*), 40 µl 2× KAPA HIFI mix (KAPA, 2602), 1.6 µl SYBR Green (25×), 0.8 µl $MgCl_2$ (0.1 M), and 12.8 µl $H_2O$. This PCR solution was placed in a thermocycler and incubated with the following parameters: 98°C for 45 s, followed by cycling of 98°C for 15 s, 60°C for 30 s, and 72°C for 60 s. Cycling continued on a qPCR machine until the reaction approached saturation. PCR products were then purified using VAHTS DNA Clean Beads at a ratio of 0.9× and eluted in 25 µl of $H_2O$. Finally, the purified PCR products underwent end repair and adaptor ligation using the VAHTS Universal DNA Library Prep Kit for Illumina V3 (Vazyme, ND607).

## Bulk RNA-seq library construction

Total RNA of the samples was extracted utilizing the Bacteria RNA Extraction Kit (R403-01, Vazyme). Subsequently, the RNA underwent mRNA enrichment (N407, Vazyme), fragmentation, cDNA synthesis, and library preparation using the VAHTSTM Total RNA-seq (H/M/R) Library Prep Kit for Illumina (NR603, Vazyme).

## Bioinformatics analysis methods

### Single-cell analysis

The sequencing data underwent processing into matrices using scripts and a pipeline as previously described (**Blattman et al., 2020**) in Python 2.7.15, with some modifications (the detailed original code and all the data were deposited in the GEO repository). After the count tables were made, subsequent analysis of single-cell data was conducted using Seurat (**Hao et al., 2021**) package (version 4.3.0; http://satijalab.org/seurat/) in R (https://www.r-project.org/). Since there were two replicates of static *E. coli* biofilm, these two datasets were merged into one SeuratObject and batch effects were removed. However, the samples for exponential period *E. coli, S. aureus,* and *C. crescentus* only had one sample, so they did not need this process. At the beginning of doing the scRNA-seq analysis, we screened the data of all samples. For preprocessing of static *E. coli* biofilm data, cells were filtered with UMI per cell more than 100 and less than 2000 for replicate 1 and replicate 2 to obtain 1621 and 3999 cells, respectively. For data of exponential period *E. coli*, the data was screened for cells with UMIs greater than 200 and less than 5000 to obtain 1464 cells. The screening criteria of *S. aureus* were cells with UMIs greater than 15 and less than 1000 and genes greater than 30 ($1000>UMIs>15$, gene counts$>30$). The screening criteria of *C. crescentus* were cells with UMIs greater than 200 and less than 5000 and gene counts greater than 30 ($5000>UMIs>200$, gene counts$>30$). After screening, all the data were normalized using a scale factor of 10,000 through a global-scaling normalization method called 'LogNormalize'. Highly variable features were then identified, returning 500 features per dataset. Then we combine the data of the two replicates of static *E. coli* biofilm into a single SeuratObject by FindIntegrationAnchors and IntegrateData functions. Then all the data underwent scaling using the ScaleData function, followed by dimension reduction through principal component analysis. To avoid subtle batch effects influencing downstream analyses, we removed batch effects using RunHarmony (**Korsunsky et al., 2019**) for the two replicates of static *E. coli* biofilm data. Then a graph-based clustering approach was employed in all data to identify clusters of gene expression programs using the Louvain algorithm (Seurat 4.3.0). The dims we chose were 6. And the resolution was 0.3 for *C. crescentus* and *S. aureus* or 0.4 for *E. coli* data. Marker genes for each cluster were computed using the Wilcoxon rank-sum test. Specifically, marker genes for each cluster were initially obtained using the FindMarkers function of Seurat. Then we performed pathway enrichment analysis of marker gene by clusterProfiler function (**Yu et al., 2012**) within R. For transcriptome-wide gene coverage across the cell population, we counted the number of genes expressing at least one UMI. Then we calculated the percentage of these genes out of all the genes in each bacterium.

### Comparison of scRNA-seq with bulk RNA-seq

The bulk RNA-seq clean data reads were mapped to the *E. coli* MG1655 k12 genome (EnsemblBacteria Taxonomy ID: 511145) using the BWA aligner software (v0.7.17-r1188, https://github.com/lh3/bwa: **Li, 2024**). Sam files were converted to bam files using samtools (v1.9). The mapping results were counted by featureCounts (https://subread.sourceforge.net/; **Liao et al., 2024**) to generate expression results. Single-cell and bulk transcriptomes of *E. coli* were compared by computing the Pearson correlation of $\log_2$ reads per gene of bulk RNA-seq and $\log_2$ UMI per gene of scRNA-seq.

### Sequencing saturation of the libraries

To assess sequencing saturation, we generated five subsamples from the single-cell sequencing data, representing 20%, 40%, 60%, 80%, and 100% of the total data. Each subsample was analyzed independently following the previously described single-cell sequencing data analysis process. We then created gene expression matrices for each subsample based on the analysis results. The number of UMIs or genes was counted for each cell detected in each subsample. Next, we sorted the cells in descending order based on their UMI counts or gene counts and selected different numbers of cells from this sorted list, starting from those with the highest UMI or gene counts. For each selection of cells, we calculated the median number of UMIs or genes. Finally, we created a line graph representing the median UMI counts and genes using GraphPad Prism 9 software, allowing us to visualize the sequencing saturation across the different subsamples.

In addition, we used the saturation calculation method of 10x Genomics to further detect the saturation of the data. The formula for calculating this metric is as follows:

Sequencing Saturation = 1 - (n_deduped_reads / n_reads). Given the differences between RiboD-PETRI and 10x Genomics datasets, we have adapted the calculation as follows:

n_deduped_reads: The number of UMIs as a measure of unique reads.

n_reads: The total number of confidently mapped reads.

## Multiplet frequency determination

Determination of the multiplet frequency was essential in assessing the efficiency of single-cell capture in RiboD-PETRI. This frequency is defined as the probability that a non-empty barcode corresponds to more than one cell. To calculate it, we used a Poisson distribution-based approach involving several key steps. Initially, we calculated the proportion of barcodes corresponding to zero cells using the formula $p(0) = \frac{\lambda^0}{0!} e^{-\lambda}$. Then, we determined the proportion for one cell, $p(1) = \frac{\lambda^1}{1!} e^{-\lambda}$, and derived the proportions for more than zero cells $p(\geq 1) = 1 - p(0)$ and more than one cell $p(\geq 2) = 1 - p(1) - p(0)$. These values allow for the calculation of the multiplet frequency as $\frac{p(\geq 2)}{p(\geq 1)}$. The parameter $\lambda$ plays a vital role in this model, representing the ratio of the number of cells to the total number of possible barcode combinations.

## Flow cytometry sorting of bacteria and analysis

All samples were measured using a Beckman CytoFLEX SRT flow cytometer with a 70 μm nozzle, using normal saline as sheath fluid. During the 24 hr static biofilm growth phase, strains labeled with BFP, PdeI-BFP, PdeI(G412S)-BFP, or c-di-GMP sensors were washed and resuspended in sterile PBS. Microorganisms were identified based on forward scatter and side scatter parameters. Cells were sorted into distinct groups according to their fluorescence intensity, with V450 used for BFP, FITC for mVenusNB, and ECD for mScarlet-I. The resulting data were subsequently analyzed using FlowJo V10 software (Tree Star, Inc).

## Antibiotic killing and persister counting assay

Cells sorted by flow cytometry were resuspended in fresh LB supplemented with 150 μg/ml ampicillin. The suspension was then incubated at 37°C for 3 hr with continuous shaking at 220 rpm. To determine the initial cell count, an aliquot of the cell suspension was taken before the ampicillin challenge, serially diluted, and plated on LB agar plates for colony-forming unit (CFU) enumeration. These plates were incubated overnight at 37°C. Following the ampicillin challenge, cells were harvested by centrifugation, washed once with sterile PBS to remove residual antibiotic, and resuspended in fresh PBS. This suspension was then serially diluted and plated on LB agar plates for post-challenge CFU counts. The persister ratio was calculated as the number of CFUs after ampicillin challenge divided by the number of CFUs before challenge. All experiments were performed in triplicate, with the results presented as means ± standard deviations of three independent biological replicates.

## Microscopy

### Bright-field and fluorescence imaging

Inverted microscopes, specifically the Nikon Eclipse Ti2 and Leica Stellaris 5 WLL, were employed for imaging, utilizing different lasers for illumination: a 405 nm laser for BFP and a 488 nm laser for GFP. Fluorescence emission signals were captured using an sCMOS camera (pco.edge 4.2 bi). Dedicated filter sets corresponding to the spectral characteristics of each fluorophore were utilized. Image analysis was performed with ImageJ software (Fiji). For the analysis of the c-di-GMP sensor, the ratio of mVenusNB to mScarlet-I (R) displayed a negative correlation with c-di-GMP concentration. Consequently, the value of $R^{-1}$ demonstrated a positive correlation with c-di-GMP concentration.

### Time-lapse imaging

To investigate the processes of antibiotic killing and bacterial resuscitation, cells labeled with PdeI-GFP during the 24 hr static growth phase were collected and washed twice with PBS. These cells were then imaged on a gel pad composed of 3% low melting temperature agarose in PBS, which was prepared as a gel island in the center of the FCS3 chamber. The cells were observed under either bright-field or epifluorescence illumination. Following the imaging, the gel pad was surrounded by LB containing

150 µg/ml ampicillin, and the cells were incubated for 6 hr at 35°C. Fresh LB was subsequently flushed in, and the growth medium was refreshed every 3 hr, ensuring sufficient recovery time for the cells.

## Determination of c-di-GMP concentration by HPLC-MS/MS

The determination of c-di-GMP concentration by HPLC-MS/MS involved a series of steps. Initially, MG1655 Δara pBAD::*pdeI* and MG1655 Δara pBAD::empty-vector strains were grown to mid-exponential growth phase, followed by induction with 0.002% arabinose. After a 2 hr incubation period, cells were harvested and washed with PBS. The washed cells were then rapidly frozen using liquid nitrogen. Simultaneously, another portion of washed cells was stained with SYTO 24 and quantified using flow cytometry. The determination of c-di-GMP concentration was conducted by Wuhan Lixinheng Technology Co. Ltd. through HPLC-MS/MS. In the experiment, first, for cell samples, addition of 500 µl of extract solvent (precooled at –20°C, acetonitrile-methanol water, 2:2:1), the samples were vortexed for 30 s, homogenized at 38 Hz for 4 min, and sonicated for 5 min in ice-water bath. The homogenate and sonicate circle were repeated for three times, followed by incubation at –20°C for 1 hr and centrifugation at 12,000 rpm and 4°C for 15 min. An 80 µl aliquot of the clear supernatant was transferred to an auto-sampler vial for LC-MS/MS analysis. The UHPLC separation was carried out using an Waters ACQUITY H-class plus UPLC System, equipped with Agilent ZORBAX Eclipse Plus C18 column (2.1 mm × 150 mm, 1.8 µm). An Waters Xevo TQ-XS triple quadrupole mass spectrometer, equipped with an electrospray ionization interface, was applied for assay development. All strains were assayed in biological triplicates, and the measured values were converted into intracellular c-di-GMP concentrations (pg) per cell.

## Quantification and statistical analysis

Statistical analysis was conducted using GraphPad Prism 9 software for Windows. The significance of results was determined using a two-tailed Student's t-test. Error bars in the data representation indicate the standard deviations of the mean from a minimum of three independent experiments. A significance threshold of $p < 0.05$ was applied throughout the analysis. To denote significant differences in the results, a system of asterisks was used: * for $p < 0.05$, ** for $p < 0.01$, *** for $p < 0.001$, and **** for $p < 0.0001$. This comprehensive approach ensured a thorough and statistically sound analysis of the c-di-GMP concentration in the studied bacterial strains.

## Acknowledgements

We thank Prof. Fan Bai (Peking University) for valuable discussions. We thank Drs. Jidong Xing and Ziyang Liu for support in bioinformatics. We also thank the members of our laboratory for helpful discussion. This work is supported by the grants to YP from the National Key R&D Program of China (2021YFC2701602), the National Natural Science Foundation of China (31970089), Science Fund for Distinguished Young Scholars of Hubei Province (2022CFA077),

Major Project of Guangzhou National Laboratory (GZNL2024A01023), and the Fundamental Research Funds for the Central Universities (2042022dx0003). This work is also supported by CG from the Natural Science Foundation of Yunnan Province of China (202001BB050005). We also thank all the staff in the Core Facilities of Medical Research Institute at Wuhan University and the Core Facilities at School of Life Sciences at Peking University for their technical support.

## Additional information

### Funding

| Funder | Grant reference number | Author |
| --- | --- | --- |
| National Natural Science Foundation of China | 31970089 | Yingying Pu |
| National Key Research and Development Program of China | 2021YFC2701602 | Yingying Pu |

| Funder | Grant reference number | Author |
|---|---|---|
| Science Fund for Distinguished Young Scholars of Hunan Province | 2022CFA077 | Yingying Pu |
| Fundamental Research Funds for the Central Universities | 2042022dx0003 | Yingying Pu |
| Major Project of Guangzhou National Laboratory | GZNL2024A01023 | Yingying Pu |
| Natural Science Foundation of Yunnan Province | 202001BB050005 | Chunming Guo |

The funders had no role in study design, data collection and interpretation, or the decision to submit the work for publication.

## Author contributions

Xiaodan Yan, Conceptualization, Resources, Data curation, Software, Formal analysis, Validation, Investigation, Visualization, Methodology, Project administration, Writing – review and editing; Hebin Liao, Conceptualization, Resources, Software, Formal analysis, Validation, Investigation, Methodology, Project administration; Chenyi Wang, Formal analysis, Validation; Chun Huang, Conceptualization, Investigation; Wei Zhang, Software, Validation; Chunming Guo, Resources, Funding acquisition; Yingying Pu, Conceptualization, Resources, Supervision, Funding acquisition, Writing – original draft, Project administration, Writing – review and editing

## Author ORCIDs

Xiaodan Yan  https://orcid.org/0009-0000-7367-1567
Yingying Pu  https://orcid.org/0000-0002-5735-8199

Reviewer #1 (Public review): https://doi.org/10.7554/eLife.97543.4.sa1
Reviewer #2 (Public review): https://doi.org/10.7554/eLife.97543.4.sa2
Author response https://doi.org/10.7554/eLife.97543.4.sa3

# Additional files

## Supplementary files

- Supplementary file 1. Primers used in this study. Related to RiboD-PETRI library construction.
- Supplementary file 2. Multiplet frequency. The specific calculation process for multiplet frequency.
- Supplementary file 3. rRNA and mRNA expression of PETRI-seq and RiboD-PETRI.
- Supplementary file 4. Various methods in rRNA depletion.
- Supplementary file 5. Sequencing information. The detailed information of RiboD-PETRI libraries.
- Supplementary file 6. The cost of RiboD-PETRI. The detailed cost breakdown of RiboD-PETRI.
- Supplementary file 7. Matrix_of_*E. coli*_3h_data_by_RiboD-PETRI_in_*Figure 1C–E*.
- Supplementary file 8. Matrix_of_*E. coli*_data_by_PETRI-seq_in_*Figure 1C*.
- Supplementary file 9. Matrix_of_*E. coli*_data_by_PETRI-seq_in_*Figure 1D*.
- Supplementary file 10. *E. coli* RNA-seq data. The result of bulk RNA-seq of exponential period *E. coli* sample in *Figure 1E*.
- Supplementary file 11. Matrix_of_Exponential_period_*E. coli*_data.
- Supplementary file 12. Matrix_of_Static_*E. coli*_biofilm-1_data.
- Supplementary file 13. Matrix_of_Static_*E. coli*_biofilm-2_data.
- Supplementary file 14. Matrix_of_SA_data.
- Supplementary file 15. Matrix_of_CC_data.
- MDAR checklist

## Data availability

Sequencing data have been deposited in GEO under accession codes GSE260458. All data generated or analysed during this study are included in the manuscript and supporting files; source data files have been provided for *Figures 1–4* and *Figure 1—figure supplement 1*, *Figure 2—figure supplement 1* and *Figure 3—figure supplement 1*.

The following dataset was generated:

| Author(s) | Year | Dataset title | Dataset URL | Database and Identifier |
|---|---|---|---|---|
| Yan X, Liao H, Wang C, Huang C, Zhang W, Guo C | 2024 | An improved bacterial single-cell RNA-seq reveals biofilm heterogeneity | https://www.ncbi.nlm.nih.gov/geo/query/acc.cgi?acc=GSE260458 | NCBI Gene Expression Omnibus, GSE260458 |

The following previously published datasets were used:

| Author(s) | Year | Dataset title | Dataset URL | Database and Identifier |
|---|---|---|---|---|
| Kuchina A, Brettner LM, Paleologu L, Roco CM, Rosenberg AB, Carignano A, Kibler R, Hirano W, William DePaolo R, Seelig G | 2020 | Microbial single-cell RNA sequencing by split-pool barcoding | https://www.ncbi.nlm.nih.gov/geo/query/acc.cgi?acc=GSE151940 | NCBI Gene Expression Omnibus, GSE151940 |
| Blattman SB, Jiang W, Oikonomou P, Tavazoie S | 2020 | Prokaryotic Single-Cell RNA Sequencing by In Situ Combinatorial Indexing | https://www.ncbi.nlm.nih.gov/geo/query/acc.cgi?acc=GSE141018 | NCBI Gene Expression Omnibus, GSE141018 |
| Wang B, Lin AE, Yuan J, Novak KE, Koch MD, Wingreen NS, Adamson B, Gitai Z | 2023 | Massively-parallel Microbial mRNA Sequencing (M3-Seq) reveals heterogenous behaviors in bacteria at single-cell resolution | https://www.ncbi.nlm.nih.gov/geo/query/acc.cgi?acc=GSE231935 | NCBI Gene Expression Omnibus, GSE231935 |
| Imdahl F, Vafadarnejad V, Homberger C, Saliba A-E, Vogel J | 2020 | Single-cell RNA-seq reports growth condition-specific global transcriptomes of individual bacteria | https://www.ncbi.nlm.nih.gov/geo/query/acc.cgi?acc=GSE119888 | NCBI Gene Expression Omnibus, GSE119888 |
| Ma P, Amemiya HM, He L, Gandhi SJ, Nicol R, Bhattacharyya RB, Smillie CS, Hung DT | 2023 | Bacterial droplet-based single-cell RNA-seq reveals antibiotic-associated heterogeneous cellular states | https://www.ncbi.nlm.nih.gov/geo/query/acc.cgi?acc=GSE180237 | NCBI Gene Expression Omnibus, GSE180237 |

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
