## [Editor Report · eLife Assessment]

This work introduces an **important** new method for depleting ribosomal RNA from bacterial single-cell RNA sequencing libraries, demonstrating its applicability for studying heterogeneity in microbial biofilms. The findings provide **convincing** evidence for a distinct subpopulation of cells at the biofilm base that upregulates PdeI expression. Future studies exploring the functional relationship between PdeI and c-di-GMP levels, along with the roles of co-expressed genes within the same cluster, could further enhance the depth and impact of these conclusions.

---

## [Referee Report · Reviewer #1 (Public review)]

Summary:

In this manuscript, Yan and colleagues introduce a modification to the previously published PETRI-seq bacterial single cell protocol to include a ribosomal depletion step based on a DNA probe set that selectively hybridizes with ribosome-derived (rRNA) cDNA fragments. They show that their modification of the PETRI-seq protocol increases the fraction of informative non-rRNA reads from ~4-10% to 54-92%. The authors apply their protocol to investigating heterogeneity in a biofilm model of *E. coli*, and convincingly show how their technology can detect minority subpopulations within a complex community.

Strengths:

The method the authors propose is a straightforward and inexpensive modification of an established split-pool single cell RNA-seq protocol that greatly increases its utility, and should be of interest to a wide community working in the field of bacterial single cell RNA-seq.

Comments on revised version:

The reviewers have responded thoughtfully and comprehensively to all of my comments. I believe the details of the protocol are now much easier to understand, and the text and methods have been significantly clarified. I have no further comments.

---

## [Referee Report · Reviewer #2 (Public review)]

Summary:

This work introduces a new method of depleting the ribosomal reads from the single-cell RNA sequencing library prepared with one of the prokaryotic scRNA-seq techniques, PETRI-seq. The advance is very useful since it allows broader access to the technology by lowering the cost of sequencing. It also allows more transcript recovery with fewer sequencing reads. The authors demonstrate the utility and performance of the method for three different model species and find a subpopulation of cells in the *E. coli* biofilm that express a protein, PdeI, which causes elevated c-di-GMP levels. These cells were shown to be in a state that promotes persister formation in response to ampicillin treatment.

Strengths:

The introduced rRNA depletion method is highly efficient, with the depletion for *E. coli* resulting in over 90% of reads containing mRNA. The method is ready to use with existing PETRI-seq libraries which is a large advantage, given that no other rRNA depletion methods were published for split-pool bacterial scRNA-seq methods. Therefore, the value of the method for the field is high. There is also evidence that a small number of cells at the bottom of a static biofilm express PdeI which is causing the elevated c-di-GMP levels that are associated with persister formation. This finding highlights the potentially complex role of PdeI in regulation of c-di-GMP levels and persister formation in microbial biofilms.

Comments on revised version:

The authors edited the manuscript thoroughly in response to the comments, including both performing new experiments and showing more data and information. Most of the major points raised between both reviewers were addressed. The authors explained the seeming contradiction between c-di-GMP levels and PdeI expression.

---

## [Author Response]

The following is the authors’ response to the previous reviews.

**eLife Assessment**
This work presents an important method for depleting ribosomal RNA from bacterial single-cell RNA sequencing libraries, enabling the study of cellular heterogeneity within microbial biofilms. The approach convincingly identifies a small subpopulation of cells at the biofilm's base with upregulated PdeI expression, offering invaluable insights into the biology of bacterial biofilms and the formation of persister cells. Further integrated analysis of gene interactions within these datasets could deepen our understanding of biofilm dynamics and resilience.

Thank you for your valuable feedback and for recognizing the importance of our method for depleting ribosomal RNA from bacterial single-cell RNA sequencing libraries. We are pleased that our approach has convincingly identified a small subpopulation of cells at the base of the biofilm with upregulated PdeI expression, providing significant insights into the biology of bacterial biofilms and the formation of persister cells.

We acknowledge your suggestion for a more comprehensive analysis of multiple genes and their interactions. While we conducted a broad analysis across the transcriptome, our decision to focus on the heterogeneously expressed gene PdeI was primarily informed by its critical role in biofilm biology. In addition to PdeI, we investigated other marker genes and noted that *lptE* and *sstT* exhibited potential associations with persister cells. However, our interaction analysis revealed that LptE and SstT did not demonstrate significant relationships with c-di-GMP and PdeI based on current knowledge. This insight led us to concentrate on PdeI, given its direct relevance to biofilm formation and its close connection to the c-di-GMP signaling pathway.

We fully agree that other marker genes may also have important regulatory roles in different aspects of biofilm dynamics. Thus, we plan to explore the expression patterns and potential functions of these genes in our future research. Specifically, we intend to conduct more extensive gene network analyses to uncover the complex regulatory mechanisms involved in biofilm formation and resilience.

**Public Reviews:**

**Reviewer #1 (Public review):**
Summary:In this manuscript, Yan and colleagues introduce a modification to the previously published PETRI-seq bacterial single cell protocol to include a ribosomal depletion step based on a DNA probe set that selectively hybridizes with ribosome-derived (rRNA) cDNA fragments. They show that their modification of the PETRI-seq protocol increases the fraction of informative non-rRNA reads from ~4-10% to 54-92%. The authors apply their protocol to investigating heterogeneity in a biofilm model of *E. coli*, and convincingly show how their technology can detect minority subpopulations within a complex community.Strengths:The method the authors propose is a straightforward and inexpensive modification of an established split-pool single cell RNA-seq protocol that greatly increases its utility, and should be of interest to a wide community working in the field of bacterial single cell RNA-seq.

We sincerely thank the reviewer for their thoughtful and positive evaluation of our work. We appreciate the recognition of our modification to the PETRI-seq bacterial single-cell RNA sequencing protocol by incorporating a ribosomal depletion step. The significant increase in the fraction of informative non-rRNA reads, as noted in the reviewer’s summary, underscores the effectiveness of our method in enhancing the utility of the PETRI-seq approach. We are also encouraged by the reviewer's acknowledgment of our ability to detect minority subpopulations within complex biofilm communities. Our team is committed to further validating and optimizing this method, and we believe that RiboD-PETRI will contribute meaningfully to the field of bacterial single-cell transcriptomics. We hope this innovative approach will facilitate new discoveries in microbial ecology and biofilm research.

**Reviewer #2 (Public review):**
Summary:This work introduces a new method of depleting the ribosomal reads from the single-cell RNA sequencing library prepared with one of the prokaryotic scRNA-seq techniques, PETRI-seq. The advance is very useful since it allows broader access to the technology by lowering the cost of sequencing. It also allows more transcript recovery with fewer sequencing reads. The authors demonstrate the utility and performance of the method for three different model species and find a subpopulation of cells in the *E. coli* biofilm that express a protein, PdeI, which causes elevated c-di-GMP levels. These cells were shown to be in a state that promotes persister formation in response to ampicillin treatment.Strengths:The introduced rRNA depletion method is highly efficient, with the depletion for *E. coli* resulting in over 90% of reads containing mRNA. The method is ready to use with existing PETRI-seq libraries which is a large advantage, given that no other rRNA depletion methods were published for split-pool bacterial scRNA-seq methods. Therefore, the value of the method for the field is high. There is also evidence that a small number of cells at the bottom of a static biofilm express PdeI which is causing the elevated c-di-GMP levels that are associated with persister formation. This finding highlights the potentially complex role of PdeI in regulation of c-di-GMP levels and persister formation in microbial biofilms.Weaknesses:Given many current methods that also introduce different techniques for ribosomal RNA depletion in bacterial single-cell RNA sequencing, it is unclear what is the place and role of RiboD-PETRI. The efficiency of rRNA depletion varies greatly between species for the majority of the available methods, so it is not easy to select the best fitting technique for a specific application.

Thank you for your insightful comments regarding the place and role of RiboD-PETRI in the landscape of ribosomal RNA depletion techniques for bacterial single-cell RNA sequencing. We appreciate the opportunity to address your concerns and clarify the significance of our method.

We acknowledge that the field of rRNA depletion in bacterial single-cell RNA sequencing is diverse, with many methods offering different approaches. We also recognize the challenge of selecting the best technique for a specific application, given the variability in rRNA depletion efficiency across species for many available methods. In light of these considerations, we believe RiboD-PETRI occupies a distinct and valuable niche in this landscape due to following reasons: (1) Low-input compatibility: Our method is specifically tailored for the low-input requirements of single-cell RNA sequencing, maintaining high efficiency even with limited starting material. This makes RiboD-PETRI particularly suitable for single-cell studies where sample quantity is often a limiting factor. (2) Equipment-free protocol: One of the unique advantages of RiboD-PETRI is that it can be conducted in any lab without the need for specialized equipment. This accessibility ensures that a wide range of researchers can implement our method, regardless of their laboratory setup. (3) Broad species coverage: Through comprehensive probe design targeting highly conserved regions of bacterial rRNA, RiboD-PETRI offers a robust solution for samples involving multiple bacterial species or complex microbial communities. This approach aims to provide consistent performance across diverse taxa, addressing the variability issue you mentioned. (4) Versatility and compatibility: RiboD-PETRI is designed to be compatible with various downstream single-cell RNA sequencing protocols, enhancing its utility in different experimental setups and research contexts.

In conclusion, RiboD-PETRI's unique combination of low-input compatibility, equipment-free protocol, broad species coverage, and versatility positions it as a robust and accessible option in the landscape of rRNA depletion methods for bacterial single-cell RNA sequencing. We are committed to further validating and improving our method to ensure its valuable contribution to the field and to provide researchers with a reliable tool for their diverse experimental needs.

Despite transcriptome-wide coverage, the authors focused on the role of a single heterogeneously expressed gene, PdeI. A more integrated analysis of multiple genes and\or interactions between them using these data could reveal more insights into the biofilm biology.

Thank you for your valuable feedback. We understand your suggestion for a more comprehensive analysis of multiple genes and their interactions. While we indeed conducted a broad analysis across the transcriptome, our decision to focus on the heterogeneously expressed gene PdeI was primarily based on its crucial role in biofilm biology. Beyond PdeI, we also conducted overexpression experiments on several other marker genes and examined their phenotypes. Notably, the lptE and sstT genes showed potential associations with persister cells. We performed an interaction analysis, which revealed that LptE and SstT did not show significant relationships with c-di-GMP and PdeI based on current knowledge. This finding led us to concentrate our attention on PdeI. Given PdeI's direct relevance to biofilm formation and its close connection to the c-di-GMP signaling pathway, we believed that an in-depth study of PdeI was most likely to reveal key biological mechanisms.

We fully agree with your point that other marker genes may play regulatory roles in different aspects. The expression patterns and potential functions of these genes will be an important direction in our future research. In our future work, we plan to conduct more extensive gene network analyses to uncover the complex regulatory mechanisms of biofilm formation.

**Author response image 1. sa3fig1:** The proportion of persister cells in the partially maker genes and empty vector control groups. Following induction of expression with 0.002% arabinose for 2 hours, a persister counting assay was conducted on the strains using 150 μg/ml ampicillin.

The authors should also present the UMIs capture metrics for RiboD-PETRI method for all cells passing initial quality filter (>=15 UMIs/cell) both in the text and in the figures. Selection of the top few cells with higher UMI count may introduce biological biases in the analysis (the top 5% of cells could represent a distinct subpopulation with very high gene expression due to a biological process). For single-cell RNA sequencing, showing the statistics for a 'top' group of cells creates confusion and inflates the perceived resolution, especially when used to compare to other methods (e.g. the parent method PETRI-seq itself).

Thank you for your valuable feedback regarding the presentation of UMI capture metrics for the RiboD-PETRI method. We appreciate your concern about potential biological biases and the importance of comprehensive data representation in single-cell RNA sequencing analysis. We have now included the UMI capture metrics for all cells passing the initial quality filter (≥15 UMIs/cell) for the RiboD-PETRI method. This information has been added to both the main text and the relevant figures, providing a more complete picture of our method's performance across the entire range of captured cells. These revisions strengthen our manuscript and provide readers with a more complete understanding of the RiboD-PETRI method in the context of single-cell RNA sequencing.

**Recommendations for the authors:**

**Reviewer #1 (Recommendations for the authors):**
The reviewers have responded thoughtfully and comprehensively to all of my comments. I believe the details of the protocol are now much easier to understand, and the text and methods have been significantly clarified. I have no further comments.
**Reviewer #2 (Recommendations for the authors):**
The authors edited the manuscript thoroughly in response to the comments, including both performing new experiments and showing more data and information. Most of the major points raised between both reviewers were addressed. The authors explained the seeming contradiction between c-di-GMP levels and PdeI expression. Despite these improvements, a few issues remain:

- Despite now depositing the data and analysis files to GEO, the access is embargoed and the reviewer token was not provided to evaluate the shared data and accessory files.

Please note that although the data and analysis files have been deposited to GEO, access is currently embargoed. To evaluate the shared data and accessory files, you will need a reviewer token, which appears to have not been provided.

To gain access, please follow these steps:

Visit the GEO accession page at: https://www.ncbi.nlm.nih.gov/geo/query/acc.cgi?acc=GSE260458

In the designated field, enter the reviewer token: ehipgqiohhcvjev

- Despite now discussing performance metrics for RiboD-PETRI method for all cells passing initial quality filter (>=15 UMIs/cell) in the text, the authors continued to also include the statistics for top 1000 cells, 5,000 cells and so on. Critically, Figure 2A-B is still showing the UMI and gene distributions per cell only for these select groups of cells. The intent to focus on these metrics is not quite clear, as selection of the top few cells with higher UMI count may introduce biological biases in the analysis (what if the top 5% of cells are unusual because they represent a distinct subpopulation with very high gene expression due to a biological process). I understand the desire to demonstrate the performance of the method by highlighting a few select 'best' cells, however, for single-cell RNA sequencing showing the statistics for a 'top' group of cells is not appropriate and creates confusion, especially when used to compare to other methods (e.g. the parent method PETRI-seq itself).

We appreciate your insightful feedback regarding our presentation of the RiboD-PETRI method's performance metrics. We acknowledge the concerns you've raised and agree that our current approach requires refinement. We have revised our analysis to prominently feature metrics for all cells that pass the initial quality filter (≥15 UMIs/cell) (Fig. 2A, Fig. 3A, Supplementary Fig. 1A, B and Supplementary Fig. 2A, G). This approach provides a more representative view of the method's performance across the entire dataset, avoiding potential biases introduced by focusing solely on top-performing cells.

We recognize that selecting only the top cells based on UMI counts can indeed introduce biological biases, as these cells may represent distinct subpopulations with unique biological processes rather than typical cellular states. To address this, we have clearly stated the potential for bias when highlighting select 'best' cells. We also provided context for why these high-performing cells are shown, explaining that they demonstrate the upper limits of the method's capabilities (lines 139). In addition, when comparing RiboD-PETRI to other methods, including the parent PETRI-seq, we ensured that comparisons are made using consistent criteria across all methods.

By implementing these changes, we aim to provide a more accurate, unbiased, and comprehensive representation of the RiboD-PETRI method's performance while maintaining scientific rigor and transparency. We appreciate your critical feedback, as it helps us improve the quality and reliability of our research presentation.

- Line 151 " The findings reveal that our sequencing saturation is 100% (Fig. S1B, C)" - I suggest the authors revisit this calculation as this parameter is typically very challenging to get above 95-96%. The sequencing saturation should be calculated from the statistics of alignment themselves, i.e. the parameter calculated by Cell Ranger as described here https://kb.10xgenomics.com/hc/en-us/articles/115003646912-How-is-sequencing-saturation-calculated :"The web_summary.html output from cellranger count includes a metric called "Sequencing Saturation". This metric quantifies the fraction of reads originating from an already-observed UMI. More specifically, this is the fraction of confidently mapped, valid cell-barcode, valid UMI reads that are non-unique (match an existing cell-barcode, UMI, gene combination).The formula for calculating this metric is as follows:Sequencing Saturation = 1 - (n_deduped_reads / n_reads)wheren_deduped_reads = Number of unique (valid cell-barcode, valid UMI, gene) combinations among confidently mapped reads.n_reads = Total number of confidently mapped, valid cell-barcode, valid UMI reads.Note that the numerator of the fraction is n_deduped_reads, not the non-unique reads that are mentioned in the definition. n_deduped_reads is a degree of uniqueness, not a degree of duplication/saturation. Therefore we take the complement of (n_deduped_reads / n_reads) to measure saturation."

We appreciate your insightful comment regarding our sequencing saturation calculation. The sequencing saturation algorithm we initially employed was based on the methodology used in the BacDrop study (PMID: PMC10014032, https://pmc.ncbi.nlm.nih.gov/articles/PMC10014032/).

We acknowledge the importance of using standardized and widely accepted methods for calculating sequencing saturation. As per your suggestion, we have recalculated our sequencing saturation using the method described by 10x Genomics. Given the differences between RiboD-PETRI and 10x Genomics datasets, we have adapted the calculation as follows:

· n_deduped_reads: We used the number of UMIs as a measure of unique reads.

· n_reads: We used the total number of confidently mapped reads.

After applying this adapted calculation method, we found that our sequencing saturation ranges from 92.16% to 93.51%. This range aligns more closely with typical expectations for sequencing saturation in single-cell RNA sequencing experiments, suggesting that we have captured a substantial portion of the transcript diversity in our samples. We also updated Figure S1 to reflect these recalculated sequencing saturation values. We will also provide a detailed description of our calculation method in the methods section to ensure transparency and reproducibility. It's important to note that this saturation calculation method was originally designed for 10× Genomics data. While we've adapted it for our study, we acknowledge that its applicability to our specific experimental setup may be limited.

We thank you for bringing this important point to our attention. This recalculation not only improves the accuracy of our reported results but also aligns our methodology more closely with established standards in the field. We believe these revisions strengthen the overall quality and reliability of our study.

- Further, this calculated saturation should be taken into account when comparing the performance of the method in terms of retrieving diverse transcripts from cells. I.e., if the RiboD-Petri dataset was subsampled to the same saturation as the original PETRI-seq dataset was obtained with, would the median UMIs/cell for all cells above filter be comparable? In other words, does rRNA depletion just decreases the cost to sequence to saturation, or does it provide UMI capture benefits at a comparable saturation?

We appreciate your insightful question regarding the comparison of method performance in terms of transcript retrieval diversity and the impact of saturation. To address your concerns, we conducted an additional analysis comparing the RiboD-PETRI and original PETRI-seq datasets at equivalent saturation levels besides our original analysis with equivalent sequencing depth.

With equivalent sequencing depth, RiboD-PETRI demonstrates a significantly enhanced Unique Molecular Identifier (UMI) counts detection rate compared to PETRI-seq alone (Fig. 1C). This method recovered approximately 20175 cells (92.6% recovery rate) with ≥ 15 UMIs per cell with a median UMI count of 42 per cell, which was significantly higher than PETRI-seq's recovery rate of 17.9% with a median UMI count of 20 per cell (Figure S1A, B), indicating the number of detected mRNA per cell increased prominently.

When we subsampled the RiboD-PETRI dataset to match the saturation level of the original PETRI-seq dataset (i.e., equalizing the n_deduped_reads/n_reads ratio), we found that the median UMIs/cell for all cells above the filter threshold was higher in the RiboD-PETRI dataset compared to the original PETRI-seq (as shown in Author response image 2). This observation can be primarily attributed to the introduction of the rRNA depletion step in the RiboD-PETRI method. Our analysis suggests that rRNA depletion not only reduces the cost of sequencing to saturation but also provides additional benefits in UMI capture efficiency at comparable saturation levels.The rRNA depletion step effectively reduces the proportion of rRNA-derived reads in the sequencing output. Consequently, at equivalent saturation levels, this leads to a relative increase in the number of n_deduped_reads corresponding to mRNA transcripts. This shift in read composition enhances the capture of informative UMIs, resulting in improved transcript diversity and detection.

In conclusion, our findings indicate that the rRNA depletion step in RiboD-PETRI offers dual advantages: it decreases the cost to sequence to saturation and provides enhanced UMI capture benefits at comparable saturation levels, ultimately leading to more efficient and informative single-cell transcriptome profiling.

**Author response image 2. sa3fig2:** At almost the same sequencing saturation (64% and 67%), the number of cells exceeding the screening criteria (≥15 UMIs) and the median number of UMIs in cells in Ribod-PETRI and PETRI-seq data of exponential period *E. coli* (3h).

- smRandom-seq and BaSSSh-seq need to also be discussed since these newer methods are also demonstrating rRNA depletion techniques. (https://doi.org/10.1038/s41467-023-40137-9 and https://doi.org/10.1101/2024.06.28.601229)

Thank you for your valuable feedback. We appreciate the opportunity to discuss our method, RiboD-PETRI, in the context of other recent advances in bacterial RNA sequencing techniques, particularly smRandom-seq and BaSSSh-seq.

RiboD-PETRI employs a Ribosomal RNA-derived cDNA Depletion (RiboD) protocol. This method uses probe primers that span all regions of the bacterial rRNA sequence, with the 3'-end complementary to rRNA-derived cDNA and the 5'-end complementary to a biotin-labeled universal primer. After hybridization, Streptavidin magnetic beads are used to eliminate the hybridized rRNA-derived cDNA, leaving mRNA-derived cDNA in the supernatant. smRandom-seq utilizes a CRISPR-based rRNA depletion technique. This method is designed for high-throughput single-microbe RNA sequencing and has been shown to reduce the rRNA proportion from 83% to 32%, effectively increasing the mRNA proportion four times (from 16% to 63%). While specific details about BaSSSh-seq's rRNA depletion technique are not provided in the available information, it is described as employing a rational probe design for efficient rRNA depletion. This technique aims to minimize the loss of mRNA during the depletion process, ensuring a more accurate representation of the transcriptome.

RiboD-PETRI demonstrates significant enhancement in rRNA-derived cDNA depletion across both gram-negative and gram-positive bacterial species. It increases the mRNA ratio from 8.2% to 81% for *E. coli* in exponential phase, from 10% to 92% for *S. aureus* in stationary phase, and from 3.9% to 54% for C. crescentus in exponential phase. smRandom-seq shows high species specificity (99%), a minor doublet rate (1.6%), and a reduced rRNA percentage (32%). These metrics indicate its efficiency in single-microbe RNA sequencing. While specific performance metrics for BaSSSh-seq are not provided in the available information, its rational probe design approach suggests a focus on maintaining mRNA integrity during the depletion process.

RiboD-PETRI is described as a cost-effective ($0.0049 per cell), equipment-free, and high-throughput solution for bacterial scRNA-seq. This makes it an attractive option for researchers with budget constraints. While specific cost information is not provided, the efficiency of smRandom-seq is noted to be affected by the overwhelming quantity of rRNAs (>80% of mapped reads). The CRISPR-based depletion technique likely adds to the complexity and cost of the method. Cost and accessibility information for BaSSSh-seq is not provided in the available data, making a direct comparison difficult.

All three methods represent significant advancements in bacterial RNA sequencing, each offering unique approaches to the challenge of rRNA depletion. RiboD-PETRI stands out for its cost-effectiveness and demonstrated success in complex systems like biofilms. Its ability to significantly increase mRNA ratios across different bacterial species and growth phases is particularly noteworthy. smRandom-seq's CRISPR-based approach offers high specificity and efficiency, which could be advantageous in certain research contexts, particularly where single-microbe resolution is crucial. However, the complexity of the CRISPR system might impact its accessibility and cost-effectiveness. BaSSSh-seq's focus on minimizing mRNA loss during depletion could be beneficial for studies requiring highly accurate transcriptome representations, although more detailed performance data would be needed for a comprehensive comparison. The choice between these methods would depend on specific research needs. RiboD-PETRI's cost-effectiveness and proven application in biofilm studies make it particularly suitable for complex bacterial community analyses. smRandom-seq might be preferred for studies requiring high-throughput single-cell resolution. BaSSSh-seq could be the method of choice when preserving the integrity of the mRNA profile is paramount.

In conclusion, while all three methods offer valuable solutions for rRNA depletion in bacterial RNA sequencing, RiboD-PETRI's combination of efficiency, cost-effectiveness, and demonstrated application in complex biological systems positions it as a highly competitive option in the field of bacterial transcriptomics.

We have revised our discussion in the manuscript according to the above analysis (lines 116-119)

- Ctrl and Delta-Delta abbreviations are used in main text but not defined there (lines 107-110).

Thank you for your valuable feedback. We have now defined the abbreviations "Ctrl" and "Delta-Delta" in the main text for clarity.

- The utility of Figs 2E and 3E is questionable - the same information can be conveyed in text.

Thank you for your thoughtful observation regarding Figures 2E and 3E. We appreciate your feedback and would like to address the concerns you've raised.

While we acknowledge that some of the information in these figures could be conveyed textually, we believe that their visual representation offers several advantages. Figures 2E and 3E provide a comprehensive visual overview of the pathway enrichment analysis for marker genes, which may be more easily digestible than a textual description. This analysis was conducted in response to another reviewer's request, demonstrating our commitment to addressing diverse perspectives in our research.

These figures allow for a systematic interpretation of gene expression data, revealing complex interactions between genes and their involvement in biological pathways that might be less apparent in a text-only format. Visual representations can make complex data more accessible to readers with different learning styles or those who prefer graphical summaries. Additionally, including such figures is consistent with standard practices in our field, facilitating comparison with other studies. We believe that the pathway enrichment analysis results presented in these figures provide valuable insights that merit inclusion as visual elements. However, we are open to discussing alternative ways to present this information if you have specific suggestions for improvement.